# GENERAL POLICY EVALUATION AND IMPROVEMENT BY LEARNING TO IDENTIFY FEW BUT CRUCIAL STATES

## ABSTRACT

Learning to evaluate and improve policies is a core problem of Reinforcement Learning (RL). Traditional RL algorithms learn a value function defined for a single policy. A recently explored competitive alternative is to learn a single value function for many policies. Here we combine the actor-critic architecture of Parameter-Based Value Functions and the policy embedding of Policy Evaluation Networks to learn a single map from policy parameters to expected return that evaluates (and thus helps to improve) any policy represented by a deep neural network (NN). The method yields competitive experimental results. In continuous control problems with infinitely many states, our value function minimizes its prediction error by simultaneously learning a small set of 'probing states' and a mapping from actions produced in probing states to the policy's return. The method extracts crucial abstract knowledge about the environment in form of very few states sufficient to fully specify the behavior of many policies. A policy improves solely by changing actions in probing states, following the gradient of the value function's predictions. Surprisingly, it is possible to clone the behavior of a near-optimal policy in Swimmer-v3 and Hopper-v3 environments only by knowing how to act in 3 and 5 such learned states, respectively. Remarkably, our value function trained to evaluate NN policies is also invariant to changes of the policy architecture: we show that it allows for zero-shot learning of linear policies competitive with the best policy seen during training.

## 1 INTRODUCTION

Policy Evaluation and Policy Improvement are arguably the most studied problems in Reinforcement Learning. They are at the root of actor-critic methods (Konda and Tsitsiklis, 2001; Sutton, 1984; Peters and Schaal, 2008), which alternate between these two steps to iteratively estimate the performance of a policy and using this estimate to learn a better policy. In the last few years, they received a lot of attention because they have proven to be effective in visual domains (Mnih et al., 2016; Wu et al., 2017), continuous control problems (Lillicrap et al., 2015; Haarnoja et al., 2018; Fujimoto et al., 2018), and applications such as robotics (Kober et al., 2013). Several ways to estimate value functions have been proposed, ranging from Monte Carlo approaches, to Temporal Difference methods (Sutton, 1984), including the challenging off-policy scenario where the value of a policy is estimated without observing its behavior (Precup et al., 2001). A limiting feature of value functions is that they are defined for a single policy. When the policy is updated, they need to keep track of it, potentially losing useful information about old policies. By doing so, value functions typically do not capture any structure over the policy parameter space. While off-policy methods learn a single value function using data from different policies, they have no specific mechanism to generalize across policies and usually suffer for large variance (Cortes et al., 2010).

Parameter Based Value Functions (PBVFs)(Faccio et al., 2021) are a promising approach to design value functions that overcome this limitation and generalize over multiple policies. A crucial problem in the application of such value functions is choosing a suitable representation of the policy. Flattening the policy parameters as done in vanilla PBVFs is difficult to scale to larger policies. Here we present an approach that connects PBVFs and a policy embedding method called "fingerprint mechanism" by Harb et al. (2020). Using policy fingerprinting allows us to scale PBVFs to handle larger NN policies and also achieve invariance with respect to the policy architecture. Policy fingerprinting was

introduced to learn maps from policy parameters to expected return offline and prior to this work was never applied to the online RL setting.

We show in visual classification tasks and in continuous control problems that our approach can identify a small number of critical "probing states" that are highly informative of the policies performance. Our learned value function generalizes across many NN-based policies. It combines the behavior of many bad policies to learn a better policy, and is able to zero-shot learn policies with a different architecture. We compare our approach with strong baselines in continuous control tasks: our method is competitive with DDPG (Lillicrap et al., 2015) and evolutionary approaches.

## 2 BACKGROUND

We consider an agent interacting with a Markov Decision Process (MDP) Stratonovich (1960); Puterman (2014) $\mathcal{M} = (\mathcal{S}, \mathcal{A}, P, R, \gamma, \mu_0)$. The state space $\mathcal{S} \subset \mathbb{R}^{n_S}$ and the action space $\mathcal{A} \subset \mathbb{R}^{n_A}$ are assumed to be compact sub-spaces. In the MDP framework, at each time-step $t$, the agent observes a state $s_t \in \mathcal{S}$, chooses an action $a_t \in \mathcal{A}$, transitions into state $s_{t+1}$ with probability $P(s_{t+1}|s_t, a_t)$, and receives a reward $r_t = R(s_t, a_t)$. The initial state is chosen with probability $\mu_0(s)$. The agent's behavior is represented by its policy $\pi : \mathcal{S} \to \Delta\mathcal{A}$: a function assigning for each state $s$ a probability distribution over the action space. A policy is deterministic when for each state there exists an action $a$ such that $a$ is selected with probability one. Here we consider parametrized policies of the form $\pi_\theta$, where $\theta \in \Theta \subset \mathbb{R}^{n_\theta}$ are the policy parameters. The return $R_t$ is defined as the cumulative discounted reward from time-step $t$, e.g. $R_t = \sum_{k=0}^{\infty} \gamma^k R(s_{t+k+1}, a_{t+k+1})$, where $\gamma \in (0, 1]$ is the discount factor. The agent's performance is measured by the expected return (i.e. the cumulative expected discounted reward) from the initial state: $J(\theta) = \mathbb{E}_{\pi_\theta}[R_0]$. The state-value function $V^{\pi_\theta}(s) = \mathbb{E}_{\pi_\theta}[R_t|s_t = s]$ of a policy $\pi_\theta$ is defined as the expected return for being in a state $s$ and following $\pi_\theta$. Similarly, the action-value function $Q^{\pi_\theta}(s, a) = \mathbb{E}_{\pi_\theta}[R_t|s_t = s, a_t = a]$ of a policy $\pi_\theta$ is defined as the expected return for being in a state $s$, taking action $a$ and then following $\pi_\theta$. State and action value functions are related by $V^{\pi_\theta}(s) = \int_{\mathcal{A}} \pi_\theta(a|s)Q^{\pi_\theta}(s, a)\, \mathrm{d}a$. The expected return can be expressed in terms of the state and action value functions by integration over the initial state distribution:

$$J(\theta) = \int_{\mathcal{S}} \mu_0(s)V^{\pi_\theta}(s)\, \mathrm{d}s = \int_{\mathcal{S}} \mu_0(s) \int_{\mathcal{A}} \pi_\theta(a|s)Q^{\pi_\theta}(s, a)\, \mathrm{d}a\, \mathrm{d}s. \tag{1}$$

The goal of a RL agent is to find the policy parameters $\theta$ that maximize the expected return. Instead of learning a single value function for a target policy, here we try to estimate the value function of any policy and maximize it over the set of initial states.

## 3 GENERAL POLICY EVALUATION

Recent work focused on extending value functions to allow them to receive the policy parameters as input. This can potentially result in single value functions defined for any policy and methods that can perform direct search in the policy parameters. We begin by extending the state-value function, and define the parameter-based state-value function (PSVF) (Faccio et al., 2021) as the expected return for being in state $s$ and following policy $\pi_\theta$: $V(s, \theta) = \mathbb{E}[R_t|s_t = s, \theta]$. Using this new definition, we can rewrite the RL objective as $J(\theta) = \int_{\mathcal{S}} \mu_0(s)V(s, \theta)\, \mathrm{d}s$. Instead of learning $V(s, \theta)$ for each state, we focus here on the policy evaluation problem over the set of the initial states of the agent. This is equivalent to trying to model $J(\theta)$ directly as a differentiable function $V(\theta)$, which is the expectation of $V(s, \theta)$ over the initial states:

$$V(\theta) := \mathbb{E}_{s \sim \mu_0(s)}[V(s, \theta)] = \int_{\mathcal{S}} \mu_0(s)V(s, \theta)\, \mathrm{d}s = J(\pi_\theta). \tag{2}$$

$V(\theta)$ is a parameter-based start-state value function (PSSVF). We consider the undiscounted case in our setting, so $\gamma$ is set to 1 throughout the paper. Once $V(\theta)$ is learned, direct policy search can be performed by following the gradient $\nabla_\theta V(\theta)$ to update the policy parameters. This learning procedure can naturally be implemented in the actor-critic framework, where a critic value function— the PSSVF—iteratively uses the collected data to evaluate the policies seen so far, and the actor follows the critic's direction of improvement to update itself. As in vanilla PSSVF, we inject noise in the policy parameters for exploration. The PSSVF actor-critic framework is reported in Algorithm1.

---

**Algorithm 1** Actor-critic with PSSVF for $V(\theta)$

---

**Input**: Differentiable critic $V_{\mathbf{w}} : \Theta \to \mathcal{R}$ with parameters $\mathbf{w}$; deterministic or stochastic actor $\pi_\theta$ with parameters $\theta$; empty replay buffer $D$

**Output** : Learned $V_{\mathbf{w}} \approx V(\theta) \forall \theta$, learned $\pi_\theta \approx \pi_{\theta^*}$

Initialize critic and actor weights $\mathbf{w}, \theta$

**repeat**:
    Perturb policy: $\theta' = \theta + \epsilon$, with $\epsilon \sim \mathcal{N}(0, \sigma^2 I)$
    Generate an episode $s_0, a_0, r_1, s_1, a_1, r_2, \ldots, s_{T-1}, a_{T-1}, r_T$ with policy $\pi_{\theta'}$
    Compute return $r = \sum_{k=1}^{T} r_k$
    Store $(\theta', r)$ in the replay buffer $D$
    **for** many steps **do**:
        Sample a batch $B = \{(r, \theta')\}$ from $D$
        Update critic by stochastic gradient descent: $\nabla_{\mathbf{w}} \mathbb{E}_{(r,\theta') \in B}[(r - V_{\mathbf{w}}(\theta'))^2]$
    **end for**
    **for** many steps **do**:
        Update actor by gradient ascent: $\nabla_\theta V_{\mathbf{w}}(\theta)$
    **end for**
**until** convergence

---

**Policy fingerprinting (Harb et al., 2020)** While the algorithm described above is straightforward and easy to implement, feeding the policy parameters as inputs to the value function remains a challenge. Recently Harb et al. (2020) showed that a form of policy embedding can be suitable for this task. Their *policy fingerprinting* creates a lower-dimensional policy representation. It learns a set of $K$ 'probing states' $\{\tilde{s}_k\}_{k=1}^{K}$ and an evaluation function $U$—like the PSSVF. To evaluate a policy $\pi_\theta$, they first compute the 'probing actions' $\tilde{a}_k$ that the policy produces in the probing states. Then the concatenated vector of these actions is given as input to $U : \mathbb{R}^{K \times n_A} \to \mathbb{R}$. While the learned probing states remain fixed when evaluating multiple policies, the probing actions in such states depend on the policy we are evaluating. The parameters of the value function $V$ are the probing states AND the weights of the MLP $U_\phi$ that maps the 'probing actions' to the return. When the policy $\pi_\theta$ is deterministic, the probing actions for such policy are the deterministic actions $\{\tilde{a}_k = \pi_\theta(\tilde{s}_k)\}$ produced in the probing states [1].

This mechanism has an intuitive interpretation: to evaluate the behavior of an agent, the PSSVF with policy fingerprinting learns a set of situations (or states), observes how the agent acts in those situations, and then maps the agent's actions to a score. Arguably, this is also how a teacher would evaluate multiple different students by simultaneously learning which questions to ask the students and how to score the student's answers.

Therefore the parameters of the value function (probing states and evaluator function) can be learned by minimizing MSE loss $\mathcal{L}_V$ between the prediction of the value function and the observed return. Setting $w = \{\phi, \tilde{s}_1, \ldots \tilde{s}_K\}$, we retrieve the common notation of $V_w(\theta)$ for the PSSVF with fingerprint mechanism. Given a batch $B$ of data $(\pi_\theta, r) \in B$, the value function optimization problem is:

$$\min_w \mathcal{L}_V := \min_w \mathbb{E}_{(\pi_\theta, r) \in B}[(V_w(\theta) - r)^2] = \min_{\phi, \tilde{s}_1, \ldots \tilde{s}_K} \mathbb{E}_{(\pi_\theta, r) \in B}[(U_\phi([\pi_\theta(\tilde{s}_1), \ldots, \pi_\theta(\tilde{s}_K)]) - r)^2] \quad (3)$$

If the prediction of the value function is accurate, policy improvement can be achieved by changing the way a policy acts in the learned probing states in order to maximize the prediction of the value function, like in the original PSSVF.

This process connects to the same interpretation as before: a student (the policy) observes which questions the teacher asks and how the teacher evaluates the student's answers, and subsequently tries to improve in such a way to maximize the score predicted by the teacher. This iterative method is depicted in Figure 1. Note that Algorithm1 applies directly to this setting. The only distinction is that the probing states are part of the learned value function. Throughout this work, with the exception of the MNIST experiments, we consider deterministic policies.

---

[1]If the policy is stochastic, the probing actions are the parameters of the output distribution of the policy in such states (the vector of probability distribution if the action space is discrete)

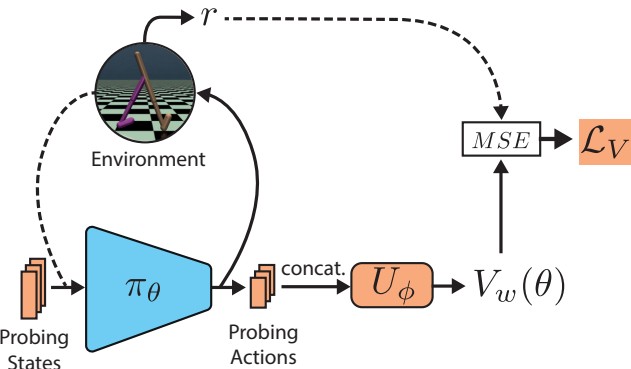

Figure 1: General policy evaluation aims to evaluate any given policy's return based on the policy's actions (referred to as probing actions) in the learned probing states. The policy can be improved through maximising the prediction of the learned value function via gradient ascent.

## 4    EXPERIMENTS

This section presents an empirical study of parameter-based value functions (PBVFs) with fingerprinting. We begin with a demonstration that fingerprinting can learn interesting states in MNIST purely through the designated evaluation task of mapping randomly initialized Convolutional Neural Networks (CNNs) to their expected loss. We also show that such a procedure could be used to construct a value function for offline improvement in MNIST. Next, we proceed to our main experiments on continuous control tasks in MuJoCo (Todorov et al., 2012). Here we show that our approach is competitive with strong baselines like DDPG (Lillicrap et al., 2015) and ARS (Mania et al., 2018), while it lacks sample efficiency when compared to SAC (Haarnoja et al., 2018). A strength of our approach is invariance to policy architecture. To illustrate this, we provide results on zero-shot learning of new policy architectures. Thereafter, we present a detailed analysis of the learned probing states in various MuJoCo environments. We conclude our study with the surprising observation that very few probing states are required to clone near-optimal behaviour in certain MuJoCo environments. An open-source implementation of our code is provided as supplementary material.

### 4.1    MOTIVATING EXPERIMENTS ON MNIST

We begin our experimental section with an intuitive demonstration of how PBVFs with fingerprinting work, using the MNIST digit classification problem. The policy is a CNN, mapping images to a probability distribution over digit classes. The environment simulation consists of running a forward pass of the CNN on a batch of data and receiving the reward, which in this case is the negative cross-entropy between the output of the CNN and the labels of the data. The value function learns to map CNN parameters to the reward (the negative loss) obtained during the simulation. Then the CNN learns to improve itself only by following the prediction of the value function, without access to the supervised learning loss. These MNIST experiments can be considered as a contextual bandit problem, where the initial state (or context) is given by the batch of training data sampled and there are no transition dynamics. We start with a randomly initialized CNN and value function and iteratively update them following Algorithm 1. Using only 10 probing states, we obtain a test set accuracy of 82.5%. When increasing the number of probing states to 50, the accuracy increases to 87%.

**Visualization of probing states**    Figure 2 shows some of the probing states learned by our model, starting from random noise. During learning, we observe the appearance of various digits (sometimes the same digit in different shapes). Since probing states are states in which the action of the policy is informative about its global behavior, it is intuitive that digits should appear. We emphasize that both the CNNs and the value function are starting from random initializations. The convolutional filters and the probing states are learned using Algorithm 1, without access to the supervised loss. For more complex datasets like CIFAR10 our method found it difficult to learn meaningful probing states.

This is possibly due to the high variance in the training data given a specific class and highlights a limitation of our method.

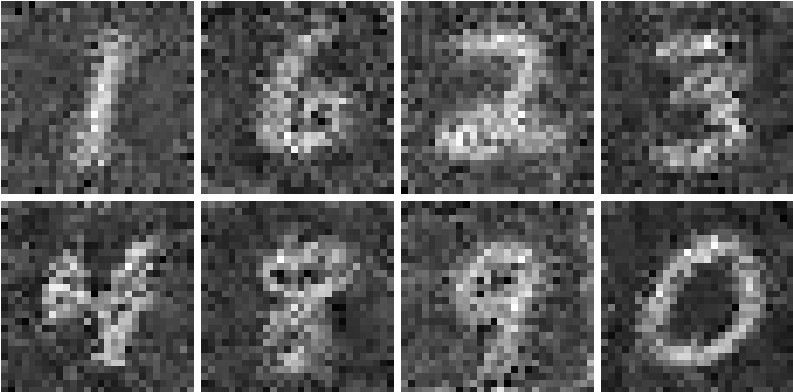

Figure 2: Samples of probing states learned while training Algorithm 1 on MNIST.

**Offline policy improvement**  Using this setting, we perform another experiment. We collect one offline dataset $\{\pi_{\theta_i}, l_i\}_{i=1}^{N}$ of $N$ randomly initialized CNN policies and their losses. We constrain the maximum accuracy of these CNNs in the training set to be 12%. We then use the dataset to train a value function offline. After training, we randomly initialize a new CNN and take many steps of gradient ascent through the fixed value function, obtaining a final CNN whose accuracy is around 65% on the test set. Our experiments show that our value function can combine the behavior of many bad NNs to produce a much better NN in a zero shot manner. We found that also with randomly initialized policies some digits appear as probing states, although they are less evident than in the online framework. We include learning curves and probing states for this scenario in Appendix B.1.

## 4.2 MAIN EXPERIMENTS ON MUJOCO

Here we present our main evaluation on selected continuous control problems from MuJoCo (Todorov et al., 2012). Since our algorithm performs direct search in parameter space, we choose *Augmented Random Search (ARS)* (Mania et al., 2018) as baseline for comparison. Moreover, since our algorithm employs deterministic policies, off-policy data, and an actor-critic architecture, a natural competitor is the *Deep Deterministic Policy Gradient (DDPG)* algorithm (Lillicrap et al., 2015), a strong baseline for continuous control. We also compare our method with the state-of-the-art *Soft Actor-Critic (SAC)* (Haarnoja et al., 2018).

**Implementation details**  For the policy architecture, we use an MLP with 2 hidden layers and 256 neurons for each layer. We use 200 probing states and later provide an analysis of them. Our implementation is based on the public code for Parameter-Based Value Functions. In some MuJoCo environments like Hopper and Walker, a bad agent can fail and the episode ends after very few time steps. This results in an excessive number of bad policies in the replay buffer, which can bias learning. Indeed, by the time a good policy is observed, it becomes difficult to use it for training when uniformly sampling experience from the replay buffer. We find that by prioritizing more recent data we are able to achieve a more uniform distribution over the buffer and increase the sample efficiency. We provide an ablation in Appendix B.2, showing the contribution of this component and of policy fingerprinting. Like in the original ARS and PBVF papers (Mania et al., 2018; Faccio et al., 2021), we use observation normalization and remove the survival bonus for the reward. The survival bonus, which provides reward 1 at each time step for remaining alive in Hopper, Walker and Ant, induces a challenging local optimum in parameter space where the agent would learn to keep still.

For DDPG and SAC, we use the default hyperparameters, yielding results on par with the best reported results for the method. For ARS, we tune for each environment step size, number of population and noise. For our method, we use a fixed set of hyperparameters, with the only exception of Ant. In Ant, we observe that setting the parameter noise for perturbations to 0.05 results in very rare positive

returns for ARS and PSSVF (after subtracting the survival bonus). Therefore we use less noise for this environment. We discuss implementation details and hyperparameters in Appendix A.

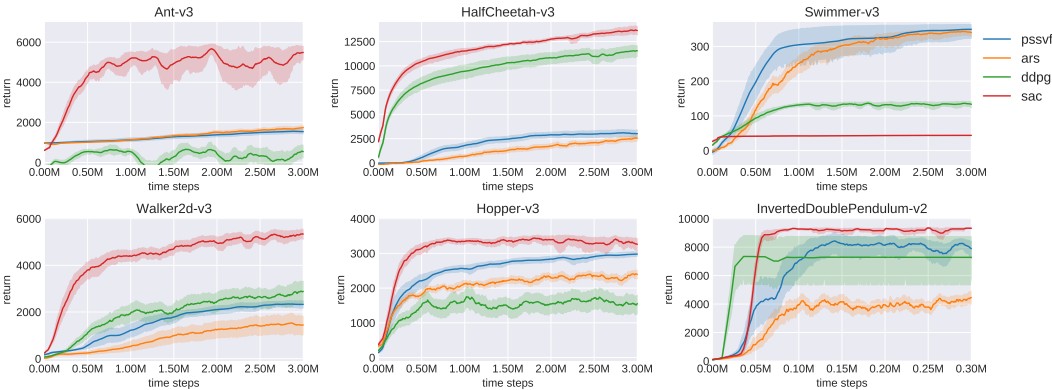

Figure 3: Return as a function of the environment interactions. The solid curve represents the mean (across 20 runs), and the shaded region represents a 95% bootstrapped confidence interval.

**Results**  Figure 3 shows learning curves in terms of expected return (mean and 95% confidence interval) achieved by our algorithm and the baselines across time in the environments. Our algorithm is very competitive with DDPG and ARS. It outperforms DDPG in all environments with the exception of HalfCheetah and Walker, and displays faster initial learning than ARS. In the Swimmer environment, DDPG and SAC fails to learn an optimal policy due to the problem of discounting[2]. On the other hand, in HalfCheetah, parameter-based methods take a long time to improve, and the ability of DDPG to give credit to sub-episodes is crucial here to learn quickly. Furthermore, the variance of our method's performance is less than DDPG's and comparable to ARS's. Like evolutionary approaches, our method uses only the return as learning data, while ignoring what happens in each state-action pairs. This is a limitation of our method and it is evident how PSSVF and ARS are less sample efficient in comparison to SAC in many environments.

In preliminary experiments we tried to learn also a function $V(s_0, \theta)$, incorporating the information on the initial state. In practice, we can store in the buffer tuples $(s_0, \theta, r)$ consisting of initial state, policy parameters and episodic return. When training the PSSVF (now similar to the PSVF), we concatenate the initial state to the probing actions and map the vector of probing actions and initial state to the return. Then policy improvement is achieved by finding the policy parameters that maximize the value function's prediction taking an expectation over the initial states sampled from the buffer. The results were very similar to those we presented in this section, so we decided to use the more straightforward approach that ignores the initial state and directly maps policy parameters to expected return. It would be also possible to learn a general value function $V(s, \theta)$ for any state, like in the PSVF algorithm (Faccio et al., 2021). We leave this as future work.

**Comparison to vanilla PSSVF**  A direct comparison to the standard Parameter-Based Value function is unfeasible for large NNs. This is because in the vanilla PSSVF, flattened policy parameters are directly fed to the value function. In our policy configuration, the flattened vector of policy parameters contains about 70K elements, which is significantly more than $200 \times n_A$ elements used to represent policies with fingerprinting. Nevertheless, we provide a direct comparison between the two approaches using a smaller policy architecture which consists of an MLP with 2 hidden layers and 64 neurons per layer. The complete results are provided in Appendix B.2. Our results in this setting show that the fingerprint mechanism could be useful even for smaller policies.

---

[2]This is a common problem for Temporal Difference methods: the policy optimizing expected return in Swimmer with $\gamma = 0.99$ is sub-optimal when considering the expected return with $\gamma = 1$. See the ablation in Appendix A.3.1 of (Faccio et al., 2021).

### 4.3 ZERO-SHOT LEARNING OF NEW POLICY ARCHITECTURES

Here we show that our method can generalize across policy architectures. We train a PSSVF using NN policies as in the main experiments. Then we randomly initialize a linear policy and start taking gradient ascent steps through the fixed value function, finding the parameters of the policy that maximizes the value function's prediction. In Figure 5 we observe that a near-optimal linear policy can be zero-shot-learned through the value function even if it was trained using policies with different architecture. It achieves an expected return of 345, while the return of best NN used for training was 360. Figure 12 (Appendix) shows results for zero-shot learning deep policies in Swimmer. We notice more variance in the performance, which might be caused by the deep policy overfitting more easily probing-state/probing-action pairs during the policy improvement phase.

### 4.4 FINGERPRINT ANALYSIS

**Ablation on number of probing states**   Our experiments show that learning probing states helps evaluating the performance of many policies, but how many of such probing states are necessary for learning? We run our main experiments again, with fewer probing states, and discover that in many environments, a very small number of states is enough to achieve good performance. In particular, we find that the PSSVF with 5 probing states achieves 314 and 2790 final return in Swimmer and Hopper respectively, while Walker needs at least 50 probing states to obtain a return above 2000. In general, 200 probing states represent a good trade-off between learning stability and computational complexity. We compare the performances of PSSVF versions with varying numbers of probing states. We use the same hyperparameters as in the main experiments (see Appendix A.2), apart for the number of probing states. Figure 4 shows that in Hopper and Swimmer 10 probing states are sufficient to learn a good policy, while Walker needs a larger number of probing states to provide stability in learning.

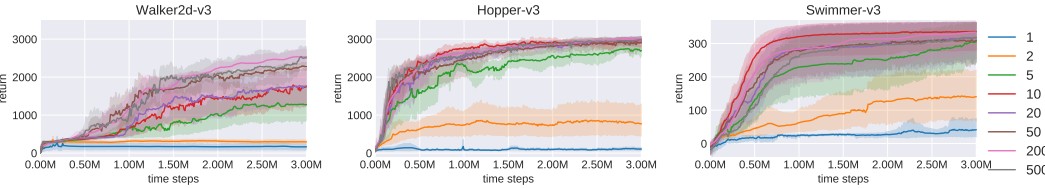

Figure 4: Average return of PSSVF with different number of probing states as a function of the number of time steps in the environment. The solid line is the average over 10 independent runs; the shading indicates 95% bootstrapped confidence intervals.

The most surprising result is that a randomly initialized policy can learn near-optimal behaviors in Swimmer and Hopper by knowing how to act only in 3 (5) such crucial learned states (out of infinitely many in the continuous state space). To verify this, we manually select 3 of the 5 learned probing states in Swimmer, and compute the actions of an optimal policy in such states. Then we train a new, randomly initialized policy, to just fit these 3 data points minimizing MSE loss. After many gradient steps, the policy obtains a return of 355, compared to the return of 364 of the optimal policy that was used to compute such actions. Figure 15 (in Appendix B.3) includes a detailed analysis of this experiment. The probing actions are the vectors $[-0.97, -0.86], [-0.18, -0.99], [0.86, 0.68]$. In the plot we notice that when the agent's state is close to the first probing state (bottom plot, depicted in blue),

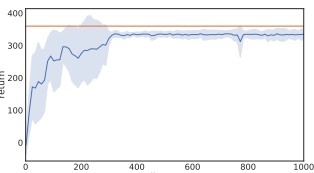

Figure 5: Performance of a linear policy (in blue) zero-shot learned (averaged over 5 runs, 95% bootstrapped CI). The orange line shows the best performance of the deep NN when training the PSSVF.

then both components of the actions are close to -1, like the probing action in such state. When the agent's state is close to the second state (bottom plot, depicted in orange), the first component

of the action moves from -1 to 0 (and then to +1) in a smooth way, while the second component jumps directly to +1. This behavior is consistent with the second probing action, since the second component is more negative than the first. Notably, although the distance between the agent's state and the third probing state (bottom plot, depicted in green) is never close to zero, such a probing state is crucial: it induces the agent to take positive actions whenever the other probing states are far away. We observe similar behavior for other environments, although they need more of such states to encode the behavior of an optimal policy. Using a similar procedure, we are able to train a randomly initialized policy in Hopper achieving 2200 return, using only 5 state-action pairs. We provide a detailed discussion and learning curves for this task in Appendix B.3.

**Visualization of RL probing states** It is possible to visualize the probing states learned by the PSSVF. To understand the behaviour in probing states, we initialize the MuJoCo environment to the learned probing state (when possible) and let it evolve for a few time steps while performing no action. In Appendix B.3 we show the crucial learned probing states of our previous experiment. Additional probing states for all environments can be seen in animated form on the website `https://anonymous260522.github.io/`.

## 5 RELATED WORK

There is a long history of RL algorithms performing direct search in parameter space or policy space. The most common approaches include evolution strategies, e.g., (Rechenberg, 1971; Sehnke et al., 2010; 2008; Wierstra et al., 2014; Salimans et al., 2017). They iteratively simulate a population of policies and use the result to estimate a direction of improvement in parameter space. Evolution strategies, however, don't reuse data: the information contained in the population is lost as soon as an update is performed, making them sample-inefficient. Several attempts have been made to reuse past data, often involving importance sampling (IS) (Zhao et al., 2013), but these methods suffer from high variance of the fitness estimator (Metelli et al., 2018). Our method directly estimates a fitness for each policy observed in the history and makes efficient reuse of past data without involving IS.

Direct search can be facilitated by compressed network search (Koutnik et al., 2010) and algorithms that distill the knowledge of an NN into another NN (Schmidhuber, 1992). Closely related to our fingerprint embedding is also the concept of Dataset Distillation (Wang et al., 2018). However, in our RL setting, learning to distill crucial states from an environment is harder due to the non-differentiability of the environment. Estimating a global objective function is common in control theory, where usually a gaussian process is maintained over the policy parameters. This allows to perform direct policy optimization during the parameter search. Such approaches are often used in the Bayesian optimization framework (Snoek et al., 2015; 2012), where a tractable posterior over the parameter space is used to drive policy improvements. Despite the soundness of these approaches, they usually employ very small control policies and scale badly with the dimension of the policy parameters. Our method, however, is invariant to policy parametrization.

It is based on a recent class of algorithms that were developed to address global estimation and improvement of policies. For Policy Evaluation Networks (PVNs) (Harb et al., 2020), an actor-critic algorithm for offline learning through policy fingerprinting was proposed. PVNs focus on the offline RL setting. In PVNs, first a dataset of randomly initialized policies with their returns is collected. Then, once their $V(\theta)$ with policy fingerprinting is trained, they perform policy improvement through gradient ascent steps on $V$. Their experimental setting is similar to our MNIST offline demonstration, which we provide just to give an intuition on how policy fingerprinting works. Concurrently, Parameter-Based Value Functions were developed to provide single value functions able to evaluate any policy, given a state, state-action pair, or a distribution over the agent's initial states. PBVFs did not use any dimensionality reductions techniques such as the policy fingerprinting mechanism, but demonstrated sample efficiency in the online RL scenario, directly using the flattened parameters of a neural network as inputs. They exhibited zero-shot learning for linear policies, but failed when the policy parameters were high-dimensional. Here, however, we demonstrated that PBVFs with policy fingerprinting mechanisms can be efficient in the online scenario. A minor difference between our approach and PVNs is that PVNs predict a discretized distribution of the return, whereas our approach simply predicts the expected return. Our method can be seen like an online version of PVN without some of the tricks used, or like a version of PSSVF where policy fingerprinting is used. Fingerprinting itself is similar to a technique for "learning to think" (Schmidhuber, 2015) where one

NN learns to send queries (sequences of activation vectors) into another NN and learns to use the answers (sequences of activation vectors) to improve its performance.

Recent work (Tang et al., 2020) learned Parameter-Based State-Value Functions which, coupled with PPO, improved performance. The authors did not use the value function to directly backpropagate gradients through the policy parameters, but only exploited the general policy evaluation properties of the method. They also proposed two dimensionality reduction techniques. The first, called *Surface Policy Representation*, consists of learning a state-action embedding that encodes possible information from a policy $\pi_\theta$. This requires feeding state-action pairs to a common MLP whose output is received as input to the value function. The MLP is trained such that it allows for both low prediction error in the value function and low reconstruction error of the action, given a state and the embedding. This method is not differentiable in the policy parameters, therefore it cannot be used for gradient-based policy improvement. The second method, called *Origin Policy Representation* (OPR), consists of using an MLP that performs layer-wise extraction of features from policy parameters. OPR uses MLPs to take as input direcly the weight matrix of each layer. This approach is almost identical to directly feeding the policy parameters to the value function (they concatenate the state to the last layer of such MLP), and suffers from the curse of dimensionality. Also, OPR was not used to directly improve the policy parameters, but only to provide better policy evaluation.

Alternative strategies to represent policies have been studied in previous work. One such strategy aims to learn a representation function mapping trajectories to a policy embedding through an auto-encoding objective (Grover et al., 2018; Raileanu et al., 2020). In particular, Grover et al. (2018) use this idea to model the agent's behavior in a multi-agent setting. The approach presented by Raileanu et al. (2020) performs gradient ascent steps finding a policy embedding that maximizes the value function's predicted return. While this maximization through the value function is similar to our setting, it relies on a representation function (or policy decoder). Our method does not use a decoder and instead directly backpropagates the gradients into the policy parameters for policy improvement. Closer to our fingerprinting setup, Pacchiano et al. (2020) utilize pairs of states and actions (from the corresponding policy) as a policy representation. However, unlike in our approach, the probing states are not learned, but sampled from a chosen probing state distribution. Kanervisto et al. (2020) suggest representing policies based on visited states via Gaussian Mixture Models applied to an offline dataset of data from multiple policies. The authors mention that their current version of policy supervectors is intended for analysing policies and is not yet suitable for online optimization. Value functions conditioned on other quantities include vector-valued adaptive critics Schmidhuber (1991), General Value Functions (Sutton et al., 2011), and Universal Value Function Approximators (Schaul et al., 2015). Unlike our approach these methods typically generalize over achieving different goals, and are not used to generalize across policies.

## 6 CONCLUSION AND FUTURE WORK

Our approach connects Parameter-Based Value Functions (PBVFs) and the fingerprinting mechanism of Policy Evaluation Networks. It can efficiently evaluate large Neural Networks, is suitable for off-policy data reuse, and competitive with existing baselines for online RL tasks. Zero-shot learning experiments on MNIST and continuous control problems demonstrated our method's generalization capabilities. Our value function is invariant to policy architecture changes, and can extract essential knowledge about a complex environment by learning a small number of situations that are important to evaluate the success of a policy. A randomly initialized policy can learn optimal behaviors in Swimmer (Hopper) by knowing how to act only in 3 (5) such crucial learned states (out of infinitely many in the continuous state space). This suggests that some of the most commonly used RL benchmarks require to learn only a few crucial state-action pairs. Our set of learned probing states is instead used to evaluate any policy, while in practice different policies may need different probing states for efficient evaluation. A natural direction for improving this method and scaling it to more complex tasks is to generate probing states in a more dynamic way, or learn to retrieve them directly from the agent's experience in the replay buffer. Like evolutionary approaches and trajectory based RL, our method might suffer high variance in stochastic environments or when the variance of the return over the initial state is high. In such scenario, poor value estimates might prevent policy improvement or zero-shot learning. Finally, PBVFs are a general framework that also considers value functions that receive states and state-action pairs as input. We plan to investigate how these value functions trained by Temporal Differences (Sutton, 1988) behave with policy fingerprinting.

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

# A    IMPLEMENTATION DETAILS

## A.1    MNIST IMPLEMENTATION

For our experiments with MNIST we adapt the official code for PSSVF to CNN policies and the MNIST classification problem.

- Policy architecture: The policy consists of two convolutional layers with 4 and 8 output channels respectively, $3 \times 3$ kernels and a stride of 1. Each convolutional layer is followed by ReLU activations. The output from the convolutional layers is flattened and provided to a fully connected linear layer which outputs the logits for the ten MNIST classes. The logits are fed into a categorical distribution; the outputs are interpreted as class probabilities.
- Value function architecture: MLP with 2 hidden layers and 64 neurons per layer with bias. ReLU activations.
- Batch size for computing the loss: 1024
- Batch size for value function optimization: 4
- Buffer size: 1000
- Loss: Cross entropy
- Initialization of probing states: uniformly random in $[-0.5, 0.5)$
- Update frequency: every time a new episode is collected
- Number of policy updates: 1
- Number of value function updates: 5
- Learning rate policy: 1e-6
- Learning rate value function: 1e-3
- Noise for policy perturbation: 0.05
- Priority sampling from replay buffer: True, with weights $1/x^{0.8}$, where $x$ is the number of episodes since the data was stored in the buffer
- Default PyTorch initialization for all networks.
- Optimizer: Adam

## A.2    RL IMPLEMENTATION

Here we report the hyperparameters used for PSSVF and the baselines. For PSSVF, we use the open source implementation provided by Faccio et al. (2021). For DDPG and SAC, we use the spinning-up RL implementation (Achiam, 2018), whose results are on par with the best reported results. For ARS, we adapt the publicly available implementation (Mania et al., 2018) to Deep NN policies.

Shared hyperparameters:

- Policy architecture: Deterministic MLP with 2 hidden layers and 256 neurons per layer with bias. Tanh activations for PSSVF and ARS. ReLu activations for DDPG and SAC. The output layer has Tanh nonlinearity and bounds the action in the action-space limit.
- Value function architecture: MLP with 2 hidden layers and 256 neurons per layer with bias. ReLU activations for PSSVF and DDPG and SAC.
- Initialization for actors and critics: Default PyTorch initialization
- Batch size: 128 for DDPG and SAC. 16 for PSSVF
- Learning rate actor: 1e-3 for DDPG and SAC; 2e-6 for PSSVF
- Learning rate critic: 1e-3 for DDPG and SAC, 5e-3 for PSSVF
- Noise for exploration: 0.05 in parameter space for PSSVF; 0.1 in action space for DDPG
- Actor's frequency of updates: every episode for PSSVF; every 50 time steps for DDPG and SAC; every batch for ARS

- Critic's frequency of updates: every episode for PSSVF; every 50 time steps for DDPG and SAC

- Replay buffer size: 100k for DDPG and SAC; 10k for PSSVF

- Optimizer: Adam for PSSVF and DDPG and SAC

- Discount factor: 0.99 for DDPG and SAC; 1 for PSSVF and ARS

- Survival reward adjustment: True for ARS and PSSVF in Hopper, Walker, Ant; False for DDPG and SAC

- Environmental interactions: 300k time steps in InvertedDoublePendulum; 3M time steps in all other environments

Tuned hyperparameters:

- Step size for ARS: tuned with values in $\{1e-2, 1e-3, 1e-4\}$

- Number of directions and elite directions for ARS: tuned with values in $\{[1, 1], [8, 4], [8, 8], [32, 4], [32, 16], [64, 8], [64, 32]\}$, where the first element denotes the number of directions and the second element the number of elite directions

- Noise for exploration in ARS: tuned with values in $\{0.1, 0.05, 0.025\}$

Hyperparameters for specific algorithms:

**PSSVF:**

- Number of probing states: 200

- Initialization of probing states: uniformly random in $[0, 1)$

- Observation normalization: True

- Number of policy updates: 5

- Number of value function updates: 5

- Priority sampling from replay buffer: True, with weights $1/x^{1.1}$, where x is the number of episodes since the data was stored in the buffer

**ARS:**

- Observation normalization: True

**DDPG and SAC:**

- Observation normalization: False

- Number of policy updates: 50

- Number of value function updates: 50

- Start-steps (random actions): 10000 time-steps

- Update after (no training): 1000 time-steps

- Polyak parameter: 0.995

- Entropy parameter (SAC): 0.2

## A.3 GPU USAGE / COMPUTATION REQUIREMENTS

Each run of PSSVF in the main experiment takes around 2.5 hours on a Tesla P100 GPU. We ran 4 instances of our algorithm for each GPU. We estimate a total of 75 node hours to reproduce our main RL results (20 independent runs for 6 environments).

## B  EXPERIMENTAL DETAILS

### B.1  MNIST EXPERIMENTS

**Online learning through Algorithm 1**    We use PSSVF (Algorithm 1) with the hyperparameters described in Appendix A.1. Figure 6 shows the performance of PSSVF using CNNs on MNIST with 10 and 50 probing states as a function of the number of interactions with the dataset. Each interaction consists of perturbing the current policy with random noise, computing the loss of the perturbed policy on a batch of data, storing the perturbed policy and its loss, and updating.

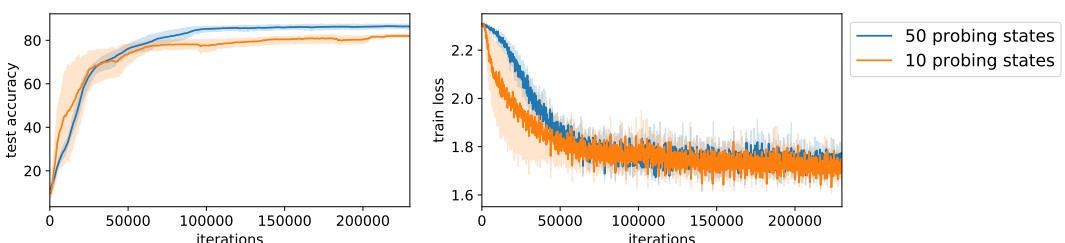

Figure 6: On the left: test accuracy of PSSVF as a function of the interactions with the dataset. On the right: loss of the perturbed CNN on the training set. Average over 5 independent runs and $95\%$ bootstrapped confidence interval.

**Visualization of learned probing states**    We plot the evolution of some of the probing states, starting from random noise, until the PSSVF is learned. We consider one run of the previous experiment with 10 probing states and show how they change during learning. This is depicted in Figure 7 where randomly initialized probing states slowly become similar to digits.

**Offline policy improvement**    This section describes the offline MNIST experiment of the main paper. Here every iteration encompasses the following steps. We perturb a randomly initialized CNN with gaussian noise with standard deviation $0.1$. Then we compute the loss on a batch of 1024 training data. If the accuracy on such batch is below $12\%$, we store the CNN and its loss, otherwise we discard the data. At every iteration we also train a PSSVF with 200 probing states, using the data collected (whose accuracy is at most $12\%$). We repeat this for 90000 iterations. Then, we randomly initialize a new CNN and train it by taking gradient steps through the fixed PSSVF, without further seeing training data. In Figure 8 we plot the performance of the zero-shot learned CNN. Surprisingly, it achieves a test accuracy of $65\%$, although only CNNs with at most $12\%$ accuracy are used in training. From the same figure we also observe that the prediction of the PSSVF is quite accurate up to 80 gradient steps, after which the performance degrades. We use a learning rate of $1e-3$ for the CNN.

**Visualization of learned probing states**    When training the PSSVF using CNNs whose accuracy is at most $12\%$, we also observe the formation of "numbers" as probing states, although they are not as evident as in the online setting. We provide some examples in Figure 9.

### B.2  MAIN EXPERIMENTS ON MUJOCO

To measure learning progress, we evaluate each algorithm for 10 episodes every 10000 time steps. We use the learned policy for PSSVF and ARS and the deterministic actor (without action noise) for DDPG. We use 20 independent instances of the same hyperparameter configuration for PSSVF and DDPG in all environments. When tuning ARS, we run 5 instances of the algorithm for each hyperparameter configuration. Then we select the best hyperparameter for each environment and carry out a further 20 independent runs. We report the best hyperparameters found for ARS in Table 1. In addition to the learning curves of the main paper in Figure 3, we report the final return with a standard deviation in Table 2.

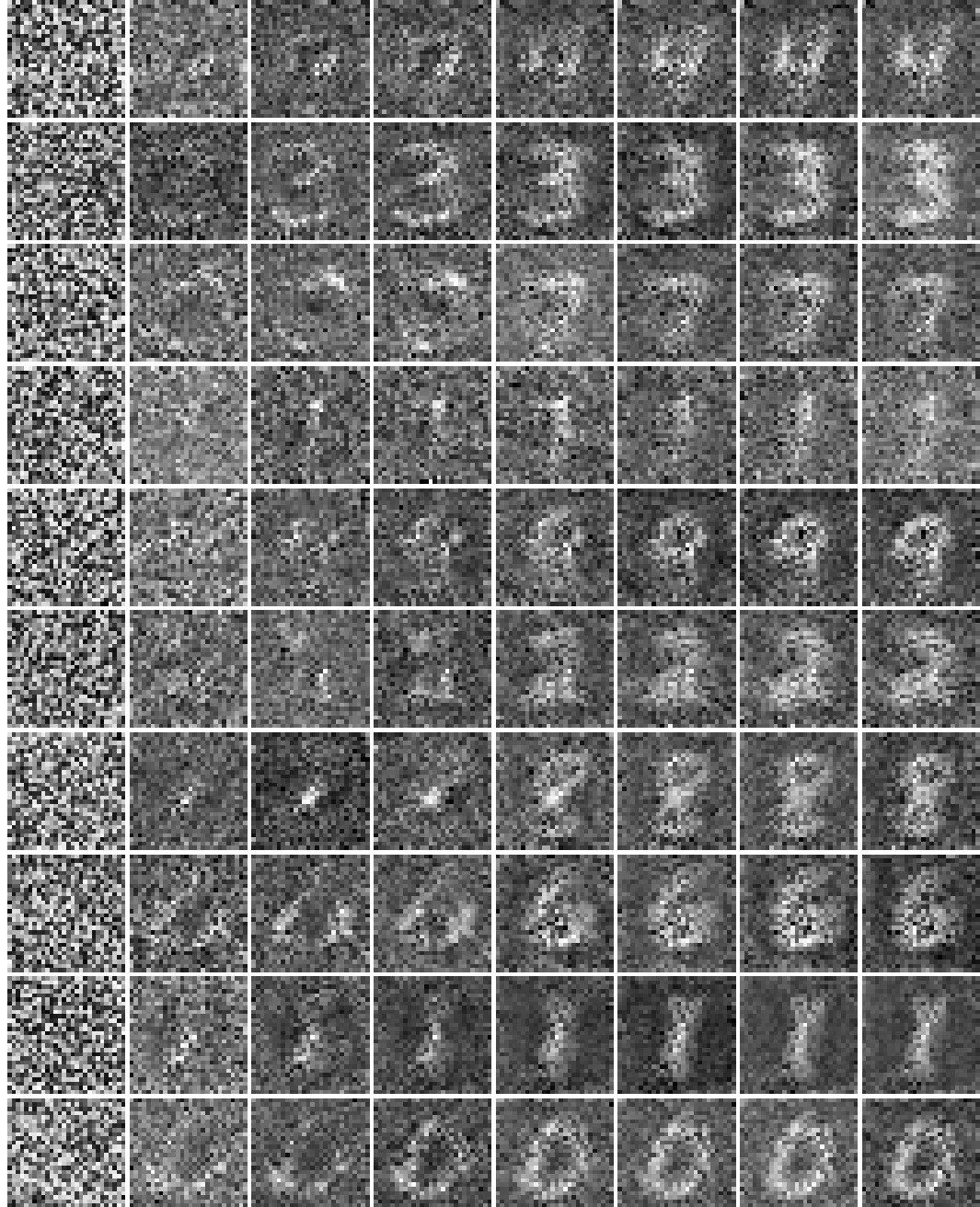

Figure 7: From left to right, the 10 probing states learned by the PSSVF using Algorithm 1. Each column represents 12500 interactions.

**Ablation on weighted sampling**    In Figure 10 we show the benefit of using non-uniform sampling from the replay buffer in Hopper and Walker environments. We compare uniform sampling (no weight) to non uniform sampling with weight $1/x^k$, where $k \in \{1.0, 1.1\}$, and $x$ is the number of episodes since the data was stored in the buffer. We achieve the best results in Hopper and Walker for the choice of $x = 1.1$. It is interesting to take this into consideration when comparing our approach to vanilla PSSVF.

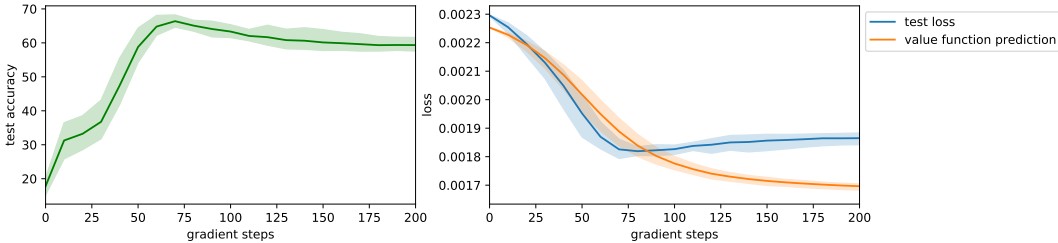

Figure 8: On the left: test accuracy of a random initialized CNN zero-shot learned using a learned PSSVF. On the right, the prediction of the performance of the CNN given by the PSSVF and the true performance on the test set. Average over 5 independent runs and $95\%$ bootstrapped c.i.

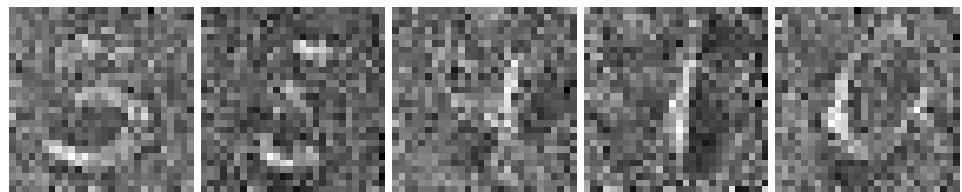

Figure 9: Samples of probing states learned by the PSSVF using CNNs with at most $12\%$ training set accuracy.

Table 1: Best hyperparameters for ARS

| Environment | step size | directions | noise |
|---|---|---|---|
| Walker2d-v3 | 0.01 | [8,8] | 0.05 |
| Swimmer-v3 | 0.01 | [8,4] | 0.05 |
| HalfCheetah-v3 | 0.01 | [8,4] | 0.05 |
| Ant-v3 | 0.01 | [32,16] | 0.01 |
| Hopper-v3 | 0.01 | [8,4] | 0.05 |
| InvertedDoublePendulum-v2 | 0.01 | [8,8] | 0.025 |

Table 2: Final return (average over final 20 evaluations)

| Environment | PSSVF | ARS | DDPG | SAC |
|---|---|---|---|---|
| Walker2d-v3 | $2333 \pm 343$ | $1488 \pm 961$ | $2432 \pm 1330$ | $\mathbf{5287 \pm 467}$ |
| Swimmer-v3 | $\mathbf{349 \pm 60}$ | $\mathbf{342 \pm 21}$ | $129 \pm 25$ | $44 \pm 1$ |
| HalfCheetah-v3 | $3067 \pm 820$ | $2497 \pm 611$ | $10695 \pm 1358$ | $\mathbf{13599 \pm 932}$ |
| Ant-v3 | $1549 \pm 240$ | $1697 \pm 225$ | $466 \pm 716$ | $\mathbf{5319 \pm 992}$ |
| Hopper-v3 | $\mathbf{2969 \pm 165}$ | $2340 \pm 199$ | $1634 \pm 1036$ | $\mathbf{3292 \pm 345}$ |
| InvertedDouble Pendulum-v2 | $\mathbf{7649 \pm 2640}$ | $4515 \pm 2733$ | $\mathbf{7377 \pm 3770}$ | $\mathbf{9235 \pm 227}$ |

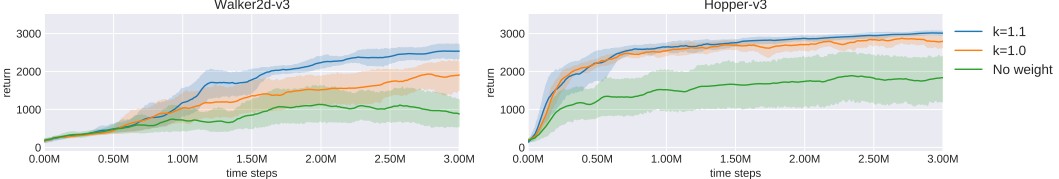

Figure 10: Comparison between our algorithm without weighted sampling from the replay buffer and with weight $1/x^k$, where $k \in \{1.0, 1.1\}$. Average over 10 independent runs and $95\%$ bootstrapped confidence interval.

**Comparison to vanilla PSSVF** Here we compare our PSSVF with policy fingerprinting to vanilla PSSVF. For vanilla PSSVF, we use the best hyperparameters reported by Faccio et al. (2021) when optimizing policies with 2 hidden layers and 64 neurons per layer and optimizing over the final rewards. Our algorithm uses the policy architecture of vanilla PSSVF and the hyperparameters of our main experiments, changing only the learning rate of the policy to $1e - 4$ and the noise for policy perturbations to $0.1$. Figure 11 shows that while in Swimmer policy fingerprinting is enough to achieve an improvement over vanilla PSSVF, in Hopper non-uniform sampling plays an important role. Note that in the vanilla PSSVF paper, learning rates and perturbation noise are tuned for each environment, while in our experiments we keep a fixed set of hyperparameters for all environments to maintain consistency. We expect the performance of our approach to also improve by selecting hyperparameters separately for each environment.

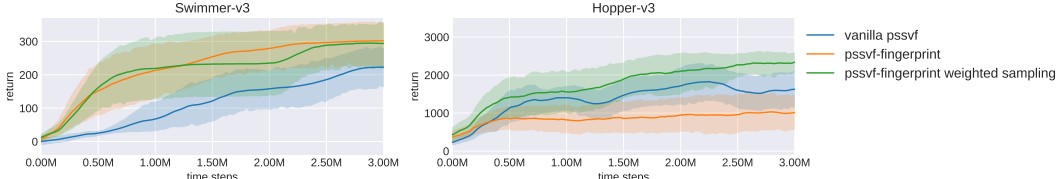

Figure 11: Comparison between vanilla PSSVF with no weighted sampling and no fingerprinting, PSSVF with policy fingerprinting, and our final algorithm that uses also weighted sampling. The solid line is the average over 10 independent runs; the shading indicates $95\%$ bootstrapped confidence intervals.

**Zero-shot learning of new policy architectures** For this task we use the same hyperparameters as in the main experiments (see Appendix A.2). We use a learning rate of $1e - 4$ to zero-shot learn the linear policy. Figure 12 reports similar results for deep policies.

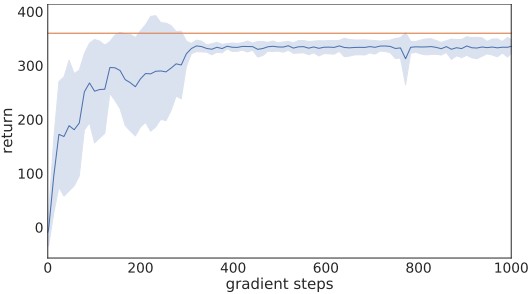

Figure 12: Performance of a deep policy (in blue) zero-shot learned (averaged over 5 runs, 95% bootstrapped CI). The orange line shows the best performance of the deep NN when training the PSSVF.

### B.3 FINGERPRINT ANALYSIS

**Learning Swimmer with 3 states** We are interested in what is the smallest amount of state-action pairs we could use to clone an optimal policy. In order to select the 3 transitions we try all combinations of 3 probing states our of 5 that we used to train our PSSVF. When cloning using all 5 probing states, the performance is very similar to the optimal policy. When choosing 4 out of 5 probing states, we notice that the performance highly depends on which probing state is removed, suggesting that some of the learned probing states are more important than others. When trying 3 out of 5 probing states this effect is more evident, and many combinations of 3 probing states lead to poor cloning performance. Here we report the learning curves for the experiment in the main paper where we fit a randomly initialized policy using only 3 transitions (see Section 4.4). These 3 transitions are 3 probing states and the corresponding optimal action (probing action) in those states. We can

see in Figure 13 that as the MSE loss goes to zero when fitting the 3 transitions, the return of the policy increases until it almost matches the optimal value. In this experiment we train a PSSVF with 5 probing states following Algorithm 1 for $2M$ time steps. We manually select a subset of 3 probing states and act in those states using the learned policy. We then fit a new policy over those 3 transition. We use a batch size of 3 and a learning rate of $2e - 5$ to fit the new policy. The other hyperparameters are the same as in the main experiments (see Appendix A.2).

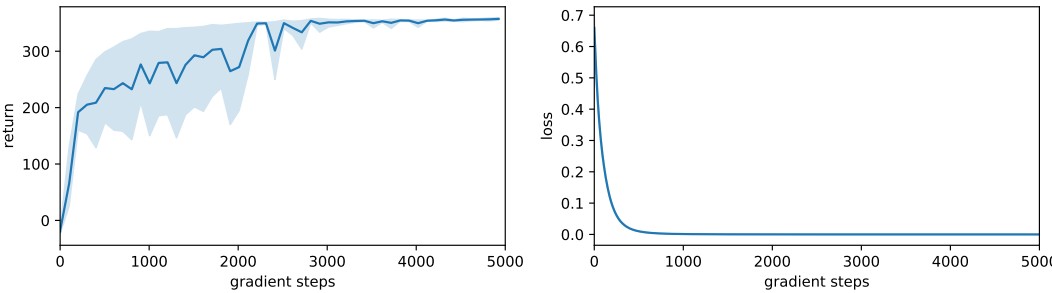

Figure 13: On the left: return of the policy learned using 3 transitions in Swimmer. On the right, MSE for fitting the 3 transitions. Average over 5 independent runs and $95\%$ bootstrapped confidence interval.

**Learning Hopper with 5 states**    We repeat the same experiment of cloning near-optimal behaviour from a few states in the Hopper environment. Using the action of a good policy (whose return is 2450) in 5 probing states, we are able to fit a new policy and obtain a final return of 2200. We use a batch size of 5 and a learning rate of $1e - 4$ for the randomly initialized policy. All other hyperparameters are like in the Swimmer experiment with 3 transitions. Figure 14 shows the learning curve, while Figure 16 relates the behavior of the policy learned using the 5 transitions to the distance of the current agent's state to the probing states. The 5 probing actions $\{\tilde{a}_k\}_{k=1}^5$ are:

$$\tilde{a}_1 = [0.4859, 0.6492, -0.7818],$$
$$\tilde{a}_2 = [0.9251, 0.9100, 0.2322],$$
$$\tilde{a}_3 = [0.0405, 0.0475, 0.9091],$$
$$\tilde{a}_4 = [0.2925, -0.4677, -0.1329],$$
$$\tilde{a}_5 = [0.7578, 0.4327, -0.1521].$$

We observe a similar behavior of the Swimmer experiments (Figure 15), where the action chosen by the agent is similar to the probing action of a probing state whenever the agent's state is close to the probing state. Although the dynamics in Hopper are more complex than in Swimmer, 5 probing states are enough to make the agent perform non-trivial actions in the environment.

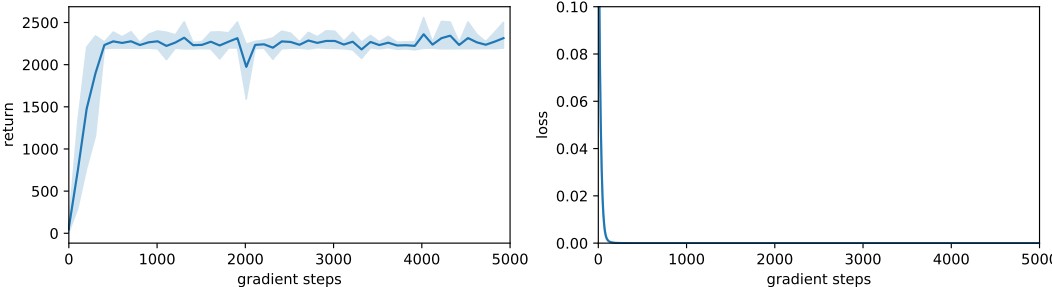

Figure 14: On the left: return of the policy learned using 5 transitions in Hopper. On the right, MSE for fitting the 5 transitions. Average over 5 independent runs and $95\%$ bootstrapped confidence interval.

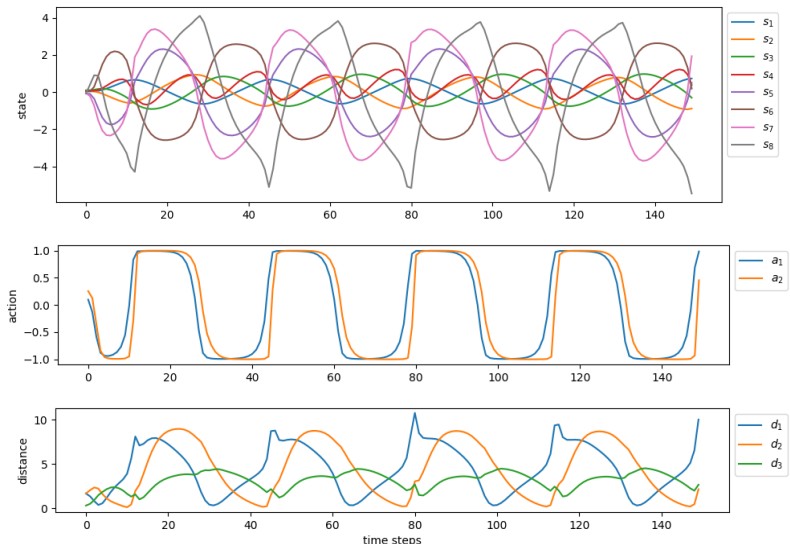

Figure 15: Behavior of the policy learned from 3 probing state-probing action pairs in Swimmer. From top to bottom: each component of the state vector across time steps in an environment simulation; each component of the action vector; L2 distance of the current state to each of the 3 probing states used for learning.

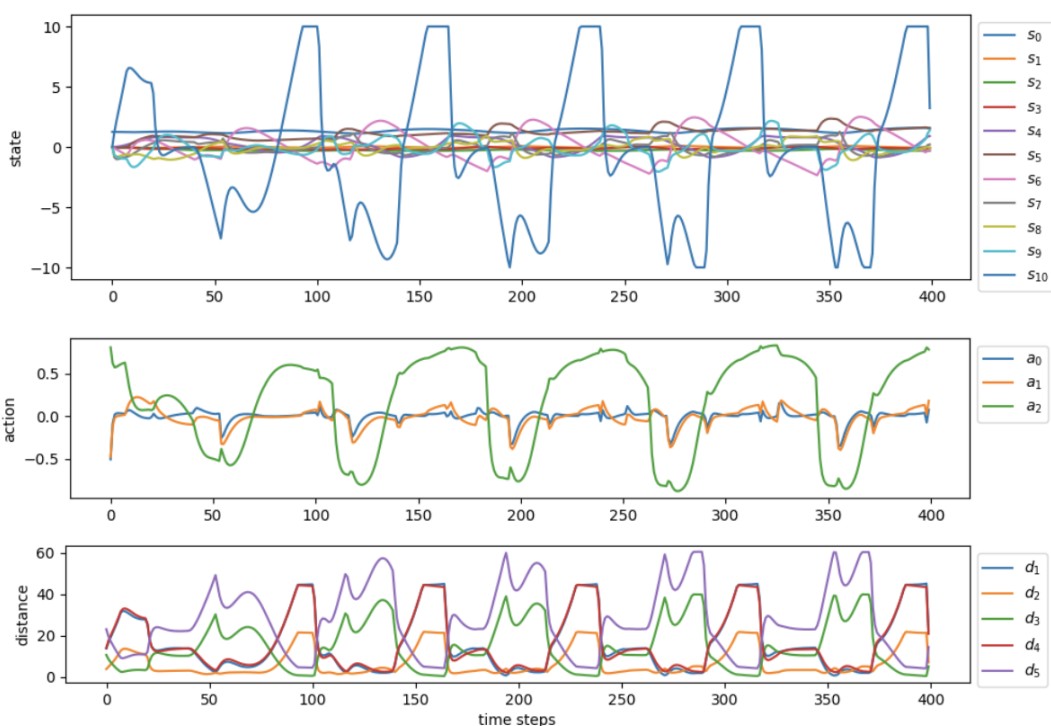

Figure 16: Behavior of the policy learned from 5 probing state-probing action pairs in Hopper. From top to bottom: each component of the state vector across time steps in an environmental simulation; each component of the action vector; L2 distance of the current state to each of the 5 probing states used for learning.

**Visualization of probing states in RL**  In Figure 15 we show the three probing states of the last experiment on Swimmer. In environments like Hopper and Walker, probing states might not correspond to a real state in the environment (e.g. some components of the probing state are outside a specific range). We notice that this is usually not the case and that the learned probing states generally correspond to valid environmental states. Moreover, we observe that probing states tend to get closer to certain critical situations over learning. These are states where certain actions have a significant effect on the future. In the Ant environment, we notice that all components of the probing state vector from index 28 to 111 learn a value of around $1e - 8$. Interestingly, the process of fingerprinting discovers this 'bug' in MuJoCo 2.0.2.2 that sets all contact forces in Ant to zero. Since these components of the state vector remain constant during the environmental interactions, and are therefore not relevant for learning, the PSSVF learns to set them to zero as well.

Figure 17 shows the evolution of the Swimmer environment from the selected probing states when no action is taken. The 3 probing states reported are those used for the experiment of Figures 13 and 15.

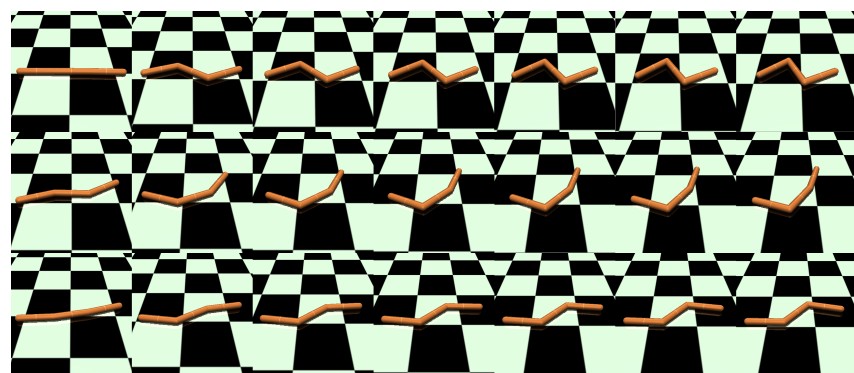

Figure 17: From top to bottom: the three learned probing states in Swimmer. From left to right: Evolution of the environment over time steps. The agent is initialized in the probing state and performs no action.

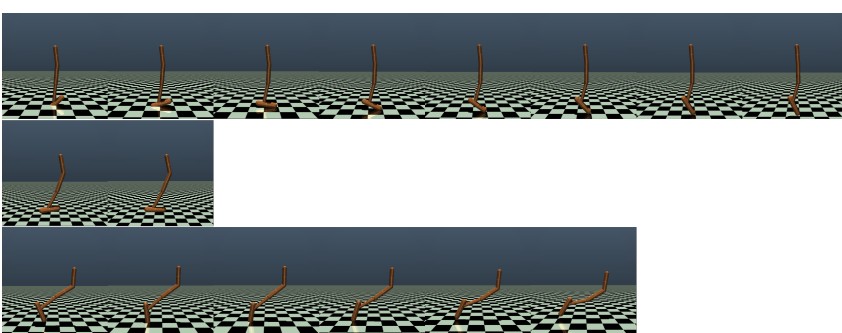

Figure 18: From top to bottom: the 5 learned probing states on Hopper. From left to right: various time steps in the environment. The agent is initialized in the probing state and performs no action.

Figure 18 shows 3 out of the 5 learned probing states on Hopper in the experiment of Figures 14 and 16. The other 2 probing states do not correspond to valid states in Hopper and are therefore not visualized. No action is taken from the probing state and the environment is allowed to evolve naturally from the probing state. The duration of interaction differs in each row of the figure as termination occurs at different points from the probing states.

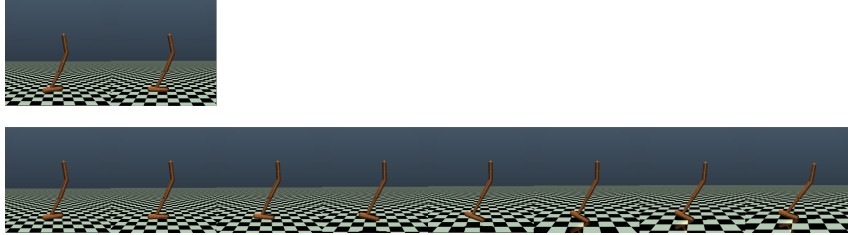

Figure 19: Evolution of the environment from a probing state when (Top) no actions taken, (Bottom) the first action in the probing state is taken using a good policy. Then no action is performed.

Figure 19 supports our hypothesis that some probing states might capture critical scenarios. In the considered probing state from Hopper we see that taking no action results in immediate failure as indicated by the shorter span of interaction in the top panel of Figure 19. In contrast, acting for a single time-step with a successful policy in that situation helps the agent survive and prolongs the interaction (bottom panel of Figure 19).

Additional probing states for all environments can be seen in animated form on the website `https://anonymous260522.github.io/`.

## C    SOCIETAL IMPACT

Our work makes algorithmic contributions to actor-critic approaches for reinforcement learning and does not focus on specific real-world applications. Using our PSSVF for offline improvement of policies (as shown in our MNIST experiment) could help mitigate risks from directly applying deep neural network policies to online situations in the real world.

## D    ENVIRONMENT DETAILS

Mujoco is made available with Apache License 2.0. The MNIST dataset is available through the creative commons license CC BY-SA 3.0.

