# OpenReview forum: "General Policy Evaluation and Improvement by Learning to Identify Few But Crucial States"
_ICLR.cc/2023/Conference — Submitted to ICLR 2023_

### Official Review · Reviewer_eb1Q · 2022-10-21

**Confidence:** 3
**Clarity, Quality, Novelty And Reproducibility:** Clearly written paper, but novelty is…
**Correctness:** 3
**Technical Novelty And Significance:** 2
**Empirical Novelty And Significance:** 2
**Recommendation:** 5

**Strength And Weaknesses:**

Strengths:
   - Interesting results : only few states are enough to get useful value functions and policy improvements in many classical settings
   - Well written paper

Weakness:
    - I feel the approach is a very straightforward extension of [Harb et al., 2020], as the only novely comes down to simply alternating the evaluation and improvement algorithms from that paper (except a very minor difference on the way of considering rewards - as a regression target rather than classification in discretized bins).
    - Thus the interest of the paper is mainly experimental. However I feel that not enough analysis is given to well understand the dynamics of the proposal.


**Summary Of The Paper:**

The paper "General Policy Evaluation and Improvement by Learning to Identify Few But Crucial States" proposes to consider fingerprints based on actions distributions on states, rather than using all policy parameters, for building value functions that generalize over all possible policies. Such policy evaluation functions are very useful for discovering efficient policies, by gradient ascent of policy parameters through the value function, as they do not require interactions to improve policies whenever the value function is general enough. Authors show interesting results.

**Summary Of The Review:**

I already gave my main concerns above, I have now some questions for the authors:
    - Is there no catastrophic forgetting during learning, since V has to embed every encountered policies ? Is there not a risk of oscillation between two parameters areas ?
    - From my point of view, the experimental part miss a study of probing states dynamics. It would be veryinsightfull to understand how they behave during learning. Do they first collapse in a single state while the value is not well learned for spreading on more useful areas afterwards ?
   - As PVN is a very related approach, I would have expected to have results of this approach as a baseline, maybe with a version that alternate evaluation and policy improvement
    - Did authors experiment a version with reward discretization as in PVN ?
   - Algorithm1 only learns from feedback over full trajectories. Thus the variance must be high. How would it perform on more difficult stochastic environments ? I suspect that considered tasks are rather simple and this kind of approach would be difficultly transferred on more complex environments.

In summary, a very easy to read and interesting paper, but from my point of view the proposal is not enough innovative for ICLR, and the experimental analysis not enough insightful.

---

> ### Author Response · Authors · 2022-11-15
> **Response to reviewer eb1Q with comments and improvements [2/2]**
>
> > From my point of view, the experimental part miss a study of probing states dynamics. It would be veryinsightfull to understand how they behave during learning. Do they first collapse in a single state while the value is not well learned for spreading on more useful areas afterwards ?
>
> We thank the reviewer for this thoughtful suggestion. Our experiments show how probing states behave during learning in the simple case of MNIST (Figure 7 in Appendix), where at the beginning of training they are quite similar to each other, but quickly become very different. In the RL setting it is more difficult to visualize how they change during training. We run our method in Swimmer using 5 probing states and plot the evolution of the 8 components of the learned states during training. As we can observe in the following plot, probing states initially have more similar values and eventually spread in different areas of the state space. In the plot, each color represents one of the 5 learned probing states.
>
> https://ibb.co/zxnsgqF
>
> If the reviewer finds these results useful, we will include them in the paper.
>
> > As PVN is a very related approach, I would have expected to have results of this approach as a baseline, maybe with a version that alternate evaluation and policy improvement - Did authors experiment a version with reward discretization as in PVN ?
>
> Since the loss used in the vanilla PBVF paper (MSE) is much more simple than the distributional loss in PVN, we opted for the former. We consider the distributional loss to be a simple trick and not a major difference between our algorithm and PVN; our method can be directly considered as an online version of PVN, we wanted to avoid the additional complexity and confounding factors introduced through discretization and the distributional loss of PVN.
>
> > Algorithm 1 only learns from feedback over full trajectories. Thus the variance must be high. How would it perform on more difficult stochastic environments ? I suspect that considered tasks are rather simple and this kind of approach would be difficultly transferred on more complex environments.
>
> This is a good point regarding the limitation of episodic RL algorithms. Our method, evolutionary approaches, and trajectory based RL, all suffer from high variance when the length of the episode is large or the environment is stochastic. Combining our method with a value function that considers also the state of the environment as input might help reducing the variance and scale-up our method to more complex domains. We further highlighted this limitation in the updated paper. Moreover, the main limitation of our method is that it learns a fixed set of probing states that are used to evaluate all policies observed. In complex stochastic environments, different policies might require different probing states for efficient evaluation. We are working on an extension of this work that resolves this issue.

---

> ### Author Response · Authors · 2022-11-15
> **Response to reviewer eb1Q with comments and improvements [1/2]**
>
> We thank the reviewer for their valuable feedback. We are glad that they found our paper well-written and our results interesting. We address the questions raised by the reviewer below.
>
> >I feel the approach is a very straightforward extension of [Harb et al., 2020], as the only novely comes down to simply alternating the evaluation and improvement algorithms from that paper (except a very minor difference on the way of considering rewards - as a regression target rather than classification in discretized bins).
>
> Although we agree that the application of network fingerprint to PSSVF might seem straightforward,
> there was no prior work showing the benefits of this approach in the online setting. Closely related
> policy representation methods have never been applied to the online setting (apart from Tang et al. [1], albeit
> the method is not used for policy improvement).
> The previous observation and our own experiences suggest that there are major challenges when
> alternating policy evaluation and policy improvement in an online framework.
> Algorithms like PVNs and PSSVF are at their earliest stages and they require many iterations
> to make them applicable to complex environments. Prior to this work, such methods were either
> unfeasible to scale-up or did not perform well even on simple tasks. We believe that our proposed approach along with a detailed analysis of the fingerprinting mechanism will be a valuable contribution to the ICLR
> community.
>
> [1] What about inputting policy in value function: Policy representation and policy-extended value function approximator. Tang et al. AAAI 2022.
>
> > Is there no catastrophic forgetting during learning, since V has to embed every encountered policies ? Is there not a risk of oscillation between two parameters areas ?
>
> We thank the reviewer for this question. By using a replay buffer, we ensure that every policy is periodically sampled and used for training. However, prioritized sampling might bias learning towards the most recent policies, which might induce some form of forgetting. Nevertheless, our method is arguably less prone to catastrophic forgetting than evolutionary methods, which do not maintain information on previously encountered policies.
>
> Note that even if we could learn the true $V(\theta)=J(\theta)$, there could be some risk of oscillation between parameters. In such a case, the learning rate for the policy update might play a crucial role in controlling oscillations.

---

### Official Review · Reviewer_zMK3 · 2022-10-21

**Confidence:** 3
**Correctness:** 2
**Technical Novelty And Significance:** 4
**Empirical Novelty And Significance:** 4
**Recommendation:** 6

**Clarity, Quality, Novelty And Reproducibility:**

see above for clarity, quality and novelty.

The experimental setup is nicely explained in the appendix, and I think reproducible.

**Strength And Weaknesses:**

I'll compress "Strength And Weaknesses" and "Clarity, Quality, Novelty And Reproducibility" into one review, as there's a lot of overlap.

### Strengths:
- the paper has a clear goal which is quite relevant: learn a single value function for evaluating any policy, and the authors clearly explain how they go about working to this goal, compare to past work on how this goal was achieved, and what they do different, and why this leads to better results.
- the paper's notation is generally clear, well introduced and easy to follow
- The experiments show what the method is doing, not just that it gets good results but that the probing states are significant, and that a value function learned even on random policies can be used to generate a good policy.
- I like that the authors don't claim their method is the best, but compare against more sample efficient methods like SAC.

### Major Concerns:

My only major concern is that the authors seem to claim to do more than they actually achieve. The claim is to "learn a single value function for evaluating...any policy" But they don't do this, they learn an estimate $J(\theta)$ of the expected return. Worse, they really dress the paper up to make it look like they learn a value function, using $V(\theta)$ notation for example, and discussing value functions and Q-functions at length in the introduction. While I'm sure their methods could be modified to learn an actual parameter-based state-value $V(s,\theta)$ I bet it's tricky (since most of the time the estimate of the value function depends on an estimate of the value function).

So I'd really wish the author's are honest with their readers and say something like "in order to focus on the main point, the probing state method and not get distracted by the complexities of learning a value function, we focus on the easier task of learning estimate $J(\theta)$ of the expected return on finite horizon MDPs. And drop anything that distracts from this simpler yet significant goal.

The authors discuss a tiny bit the step to learning a real value function at the top of page 7, but it's a bit of a farce, since they only learn the value function over initial states, if I remember correctly in MuJoCo initial states are all quite similar, so it's unsurprising that the authors state "The results were very similar to those we presented in this section"

For this reason, I have to say on the correctness that "Several of the paper’s claims are incorrect or not well-supported," but I don't see any reason why this can't be fixed.

### Minor Concerns:
- I don't find Figure 1 helpful at all. Why are some lines solid, why are some dashed? What's $U_\phi$? The algorithm has a replay buffer (or storage of policy parameters / rewards) that's not in Figure 1. Either have a helpful figure or no figure at all but not a figure for figure's sake.
- I think that Eq. 2 is incorrect, but since I think it's a distraction and should be dropped I won't go into detail.
- in eq. 4, $\pi_\theta(s_1)$ is a distribution over actions. You discussed "parameters of the output distribution of the policy in such states" but left that out in eq. 4. For eq. 4 to be correct, this mapping from $\theta$ to distribution parameterization must be included.
- Please also tell the readers that the estimation of $V(\theta)$ only works since in MuJoCo the starting states are all very similar, and the MNIST experiment is stateless.



**Summary Of The Paper:**

In this paper, the authors get the parameter-based state-value function (PSVF) Faccio et al., 2021 to work. The motivation is simple, if there's a value function $V:\mathcal S\times\Theta\to\mathbb R$ which maps both a state and the parameters of a policy to an accurate cumulative discounted reward estimate, then the policy can be improved by maximizing $V(s,\theta)$ w.r.t. $\theta$. The only problem is that for any class of policies to be useful, the parameter space of that class of polices $\Theta$ is generally quite large (thousands of dimensions) making learning the PSVF V impractical.

The authors propose to map this large $\mathcal S\times\theta$ to a smaller space with two simple tricks. First, they don't worry about the discounted return will be from a specific state, peruse a general policy evaluation framework to learn the sum of discounted rewards from an average starting state. This isn't novel. The novel and significant step they take is to learn K "probing states" $s_1,\dots,s_K$ and instead of learning the PSVF they estimate (this notation only works for deterministic policies, but the idea works for certain stochastic policies as well) the value of a policy with $V_\phi(\pi_\theta(s_1),\dots,\pi_\theta(s_K))$. Since $K$ is small (generally 200 or less) and the action space is also small, this addresses the curse of dimensionality.

The authors then do two sets of experiments, one on MNIST and the other on MuJoCo outlining interesting characteristics of their PSSVF method, showing that the probing states learned are meaningful and that the learned value estimation is useful for constructing good policies.






**Summary Of The Review:**

Overall, when reading the paper I felt I learned a significant new tool, one that I feel others in the RL community should learn

---

> ### Author Response · Authors · 2022-11-15
> **Response to reviewer zMK3 with comments and improvements**
>
>
> We thank the reviewer for their valuable feedback. We are glad that they found our goal clear and relevant, our notation easy to follow, and the use of probing states significant. Below we address the questions raised by the reviewer:
>
> > My only major concern is that the authors seem to claim to do more than they actually achieve. The claim is to "learn a single value function for evaluating...any policy" But they don't do this, they learn an estimate  of the expected return.
> ...
> For this reason, I have to say on the correctness that "Several of the paper’s claims are incorrect or not well-supported," but I don't see any reason why this can't be fixed.
>
> We thank the reviewer for pointing out this important issue. The reviewer is indeed correct and we agree that we should make clear from the beginning that we are considering only an estimate of $J(\theta)$. We used the same notation as in the PBVF and PVN papers when referring to the start-state value function $V(\theta)$. While we believe it is correct to call it a "value function", since it is a function estimating the value of a policy (in line also with previous papers on this), we now specify from the abstract that we consider a value function that simply maps policy parameters to expected return.
>
> Furthermore, following also the suggestion of Reviewer xK3x, we removed part of the background related to off-policy learning (e.g. former Eq. 2 and relative text), in favour of a more direct introduction of our method.
>
> Concerning our short discussion on $V(s_0, \theta)$ and on how to possibly train a general $V(s,\theta)$, we are happy to move this part to the Appendix or to remove it from the paper if the reviewer believes it is distracting from the main narrative.
>
> > I don't find Figure 1 helpful at all. Why are some lines solid, why are some dashed? What's $U_{\phi}$? The algorithm has a replay buffer (or storage of policy parameters / rewards) that's not in Figure 1. Either have a helpful figure or no figure at all but not a figure for figure's sake.
>
> We apologize for the confusion regarding Figure 1. Dashed lines indicate parts that are not in control of the agent (environmental state and reward), while solid lines indicate the computational steps required for evaluation of the current policy. We called $U_{\phi}$ the evaluator function that maps the concatenated probing actions to the expected return in order to distinguish it from $V_w(\theta)$, where $w$ contains the parameters $\phi$ and the probing states. Unfortunately, we did not update such a notation in the text (Eq. 4 and lines 5 and 7 of the "Policy Fingerprinting" paragraph). Thank you for pointing this out.
>
>
> > I think that Eq. 2 is incorrect, but since I think it's a distraction and should be dropped I won't go into detail.
>
> Former Eq. 2 is the standard stochastic off-policy objective. We could discuss with the reviewer whether it is a good objective for off-policy RL, but since it is now removed from the main paper and it is not useful for our results, we can agree to not further discuss it.
>
>
>
> > in eq. 4, $\pi_{\theta}(s_1)$ is a distribution over actions. You discussed "parameters of the output distribution of the policy in such states" but left that out in eq. 4. For eq. 4 to be correct, this mapping from $\theta$ to distribution parameterization must be included.
>
> We agree that Eq. 4 (now Eq.3) is only valid for deterministic policies. For stochastic policies we would have to introduce a new notation. Since our main results involve deterministic policies, in order to not confuse the reader we moved our comment on stochastic policies as a footnote.
>
> > Please also tell the readers that the estimation of $V(\theta)$ only works since in MuJoCo the starting states are all very similar, and the MNIST experiment is stateless.
>
> We agree with the reviewer on this point. Rather than the initial state being different, what matters most is the variance of the return of the agent, given the distribution over the initial state. If such variance is high, then our algorithm might suffer. This is related to the last point made by Reviewer eb1Q and our method suffer it with the same degree as evolutionary methods or trajectory-based RL methods. We further highlighted this limitation in the conclusions.
>
> Please note that the MNIST experiments, despite not having transition dynamics, contain some variance over the initial state. Here the initial state is given by the batch of images sampled from the training dataset. We specified this in the updated version of the paper.

---

> ### Author Response · Authors · 2022-12-09
> **Friendly reminder**
>
> The discussion period will end soon and we are happy to provide additional comments and engage in further discussion if needed.
> If the reviewer is satisfied with our response, we would appreciate it if they would consider increasing the score for the submission.

---

### Official Review · Reviewer_yJcN · 2022-10-23

**Confidence:** 4
**Correctness:** 4
**Technical Novelty And Significance:** 2
**Empirical Novelty And Significance:** 3
**Recommendation:** 5

**Clarity, Quality, Novelty And Reproducibility:**

# Detailed Comments (Clarity, Quality, Novetly, Reproducibility)

-   Section 2 (Stochastic vs deterministic): Relatively minor but, I do
    not think these two cases should be treated separately, as the
    objective written in equation 1 is still the objective being
    maximized. The policy-gradient theorem merely provides an
    alternative update for the policy directly. The deterministic
    objective can still be thought of as an integral, where the
    distribution is a dirac delta on the deterministic action.

-   Section 3 (Probing states and policy visitation distributions): The
    idea of having a single value function for many policies is
    appealing. The parameter representation is a global description of
    the policy. But, the probing states are local and depend on the
    policy and the visitiation distribution it induces. For example, two
    deterministic policies travelling in separate directions will never
    visit the same states, and hence the probing states may not be
    shareable amongst these two policies.

    Is there a theoretical justification or intuition for evaluating all
    the policies on one distribution of probing states (even if these
    states are learned)?

-   Section 3 (Policy Improvement and Probing states): Because the
    probing states are learned, it may be the case that the probing
    states do not correspond to any real state in the environment.
    Improvements on these probing states benefit the policy globally
    only through generalization. Returning to the "two direction MDP"
    example from my earlier point, the action on the learned probing
    state could be sufficient for evaluation/prediction and yet may not
    provide a useful gradient.

-   Section 4.1 (Offline policy improvement): It is quite surprising
    that, after constraining the loss to be 12\%, the value function is
    able to extrapolate and provide a gradient that improves the CNN to
    65\%. MNIST, however, is a relatively simple dataset which can be
    classified with a linear decision boundary. I wonder if similar
    results hold on harder datsets, such as omniglot, fashion MNIST and
    CIFAR.

# Minor Comments

-   Section 2: Markovianity -> Markov Property
-   Section 4.3: "trough" -> through



**Strength And Weaknesses:**

# Strengths

-   The main result is very interesting and surprising. The result being
    that policies in mujoco environments can be represented by observing
    the actions at a small number of states. It is further surprising
    that a strong policy can be learned by regressing on the actions at
    those few states.

-   While there are a few clarity issues, which I outline in the
    detailed comments below, the overall paper is well-structured and
    easy-to-follow. Not only is the motivation clear, the contribution
    is clear as well.

# Weaknesses

-   The main weakness is that the novelty is incremental.
    Algorithmically, there is little novelty: the proposed algorithm is
    a combination of the architecture of policy evaluation networks and
    online algorithm structure of parameter-based value functions. The
    eponymous finding, that policy evaluation and improvement can be
    done by identifying few but crucial states, is limited to only two
    mujoco environments. While the experiments encompass more than this
    finding, this seems to be the major contribution.


**Summary Of The Paper:**


This paper proposes using the policy evaluation networks architecture
(policy fingerprinting) with the algorithm structure of parameter-based
value functions to learn a single value of the start state that
conditions on the policy representation. This value function is learned
through online interaction, like parameter-based value functions, but
scales to larger policies because of policy fingerprinting. The
eponymous finding is that, for some mujoco environments, the number of
probing states for policy fingerprinting is surprisingly low while
facilitating zero-shot policy learning on different architectures.



**Summary Of The Review:**

It is difficult to rate this submission because, while lacking in
novelty, the eponymous finding is rather surprising and interesting. The
lack of novelty in the algorithm is unavoidable, as it builds on two
distinct works that are quite similar. However, I think the experiments
have potential to provide more insight. While the results are mostly
positive, the submission could be strengthened by outlining the limits
the approach in negative results. For example, is it more or less
difficult to learn to predict the losses when the dataset is more
complicated (omniglot, fashion MNIST, CIFAR). Does this difficulty
translate to difficulty to extrapolate to new architectures, or to
zero-shot train an architecture to a performance level beyond what was
seen? There is plenty of opportunity to contextualize the very
interesting main finding. As the submission currently stands, I am
leaning towards weak reject. I am open to increasing my score of some of
these points are addressed.

---

> ### Author Response · Authors · 2022-11-15
> **Response to reviewer yJcN with comments and improvements [2/2]**
>
> > Section 4.1 (Offline policy improvement): It is quite surprising that, after constraining the loss to be 12\%, the value function is able to extrapolate and provide a gradient that improves the CNN to 65\%. MNIST, however, is a relatively simple dataset which can be classified with a linear decision boundary. I wonder if similar results hold on harder datsets, such as omniglot, fashion MNIST and CIFAR.
>
>
> We are glad the reviewer finds these results suprising. MNIST is a very simple dataset, but it gives us the opportunity to properly visualize probing states and test the generalization capabilities of out method. Before submitting the paper we tried to run our Algorithm 1 (online, starting from randomly initialized CNN and value function) on CIFAR10 dataset and achieved a maximum test accuracy of only 33\%. The resulting probing states are very blurred images, which help only capturing simple features like the position of the sky or similar. Since the main focus of our experiments was RL, we did not try to extensively tune our algorithm on CIFAR. If the reviewer finds these results useful, we will add them to the Appendix. However, given the high variability of the data given even a single class, the application of our method is quite limited for such more complex datasets.
>
> > While the results are mostly positive, the submission could be strengthened by outlining the limits the approach in negative results. For example, is it more or less difficult to learn to predict the losses when the dataset is more complicated (omniglot, fashion MNIST, CIFAR). Does this difficulty translate to difficulty to extrapolate to new architectures, or to zero-shot train an architecture to a performance level beyond what was seen? There is plenty of opportunity to contextualize the very interesting main finding.
>
> We thank the reviewer for this suggestion. In the updated paper we expanded our discussion on the following. What our method is trying to solve (i.e. distilling knowledge from the dataset or environment given a scalar performance) is a very complex inverse problem. For complex datasets we believe it can be hard to learn a precise probing state, partially because of the difficulties of exploring in parameter space. Moreover, for environments with dynamics (RL), the main limitation is that our method is trying to learn the same set of probing states to evaluate very different policies. In practice, different policies might require different probing states for efficient exploration.
>
> In other words, for complex datasets there might be either an exploration problem or a fitting problem. If any of the two happen, it is very hard for our method to find good improving directions for the gradient of the policy parameters, preventing policy improvement or zero-shot learning. Zero-shot learning might be related to catastrophic forgetting.
>
> Regarding transferring to different architectures, the performance mostly depends on the capacity of the policy and on how much the value function overfits the data.
>
> > I am open to increasing my score of some of these points are addressed.
>
> We thank the reviewer again and hope that they will consider increasing their score given our response.

---

> ### Author Response · Authors · 2022-11-15
> **Response to reviewer yJcN with comments and improvements [1/2]**
>
> We thank the reviewer for their valuable feedback and for finding our results interesting and surprising and our paper well-structured and easy-to-follow. Below we address the questions raised by the reviewer:
>
> > The main weakness is that the novelty is incremental. Algorithmically, there is little novelty: the proposed algorithm is a combination of the architecture of policy evaluation networks and online algorithm structure of parameter-based value functions.
>
> We agree that the two main components of are algorithm are of limited novelty. However, we would like to stress that their combination is not trivial. This combination allows us to overcome limitations of previous approaches to learn a value function which can evaluate many policies. Importantly, our method scales to more complex environments and policies compared to those considered in the PVN and PBVF papers. Unlike PVN, our approach is also shown to perform well in online settings.
>
> > Section 2 (Stochastic vs deterministic): Relatively minor but, I do not think these two cases should be treated separately, as the objective written in equation 1 is still the objective being maximized. The policy-gradient theorem merely provides an alternative update for the policy directly. The deterministic objective can still be thought of as an integral, where the distribution is a dirac delta on the deterministic action.
>
> We agree with that reviewer that such distinction can be avoided. Moreover, since that part is not crucial for the illustration of our method, we followed the suggestion of the other reviewers and removed it from the paper in favor of more insightful contextualization of our method.
>
> > Section 3 (Probing states and policy visitation distributions): The idea of having a single value function for many policies is appealing. The parameter representation is a global description of the policy. But, the probing states are local and depend on the policy and the visitiation distribution it induces. For example, two deterministic policies travelling in separate directions will never visit the same states, and hence the probing states may not be shareable amongst these two policies. Is there a theoretical justification or intuition for evaluating all the policies on one distribution of probing states (even if these states are learned)?
>
>
> This is a very good point and, as we mention in the conclusion, it is perhaps the main current limitation of our method: different policies might require different probing states for efficient evaluation. We are currently working on an extension of this approach which learns probing states in a more dynamic way.
>
> > Section 3 (Policy Improvement and Probing states): Because the probing states are learned, it may be the case that the probing states do not correspond to any real state in the environment. Improvements on these probing states benefit the policy globally only through generalization. Returning to the "two direction MDP" example from my earlier point, the action on the learned probing state could be sufficient for evaluation/prediction and yet may not provide a useful gradient.
>
> We agree that it is difficult to estimate the relative performance of each probing state for policy improvement, especially when the probing state does not correspond to a true environmental state. In our experiment cloning a good policy (which only involves policy improvement steps) in Hopper using 5 probing states, we noticed that 2 of the learned probing states do not correspond to true environmental states. Removing any of those 2 states from the dataset resulted in much worse cloning performance, suggesting that they are still very helpful for generalization: they might correspond to situations that the agent must avoid. When the agent gets closer to such non-physical states, the probing action should move away from those situations.

---

> ### Author Response · Authors · 2022-12-09
> **Friendly reminder**
>
> The discussion period will end soon and we are happy to provide additional comments and engage in further discussion if needed.
> If the reviewer is satisfied with our response, we would appreciate it if they would consider increasing the score for the submission.

---

### Official Review · Reviewer_xK3x · 2022-10-24

**Confidence:** 4
**Correctness:** 2
**Technical Novelty And Significance:** 3
**Empirical Novelty And Significance:** 2
**Recommendation:** 3

**Clarity, Quality, Novelty And Reproducibility:**

The combination of methods is novel, but the utility is not demonstrated.

In addition, the experiments are oddly limited:
- only continuous control problems are used for main experiments
- only deterministic policies are used for main experiments
- only environments where few crucial states are needed are used for evaluation
To support the claim that probing states are a better representation, thorough comparisons between different representations should be performed.
In particular, to demonstrate that probing states scale better, evaluations should be performed with more complex environments.

The prioritization of content inclusion needs to be improved. Key experiments are in the appendix while lots of content in the main body can be cut or moved to the appendix.
Note that "reviewers are not required to read the appendix" (ICLR 2023 CFP). I have still read the appendix, but I urge the authors to move key experiments to the main body.
Examples:
- Sec 2 and 3 contain lots of non-novel content that is not needed within this work.
- Likewise, "a demonstration that fingerprinting can learn interesting states in MNIST" is of limited interest (no hypothesis is being tested; no baselines are used for comparison).
- Also, learning a near-optimal linear policy does not show that this method has an advantage over directly using policy parameters (i.e., this ability is due to PBVF).
- Evaluating the contribution of policy fingerprinting is crucial, yet it is in the appendix.

**Strength And Weaknesses:**

This work identifies a promising combination of two existing methods.
If probing states are a more efficient representation for complex policies, then this method is of interest.

The authors claim that "Flattening the policy parameters ... is difficult to scale to larger policies," but do not show the new method scaling.
Unfortunately, the proposed method is not shown to scale to larger domains. Empirical evaluation is required to confirm favorable scaling, but all of the domains considered can be solved with simple policies.
I am skeptical that this method scales better than using the policy parameters: a strength of policy gradient methods is that the policy complexity generally scales more slowly than the environment complexity. However, this method requires coverage of the state space, so the number of required crucial states may scale quickly.
The key experiment is shown in the appendix (Fig 10) and is quite limited in scope. It specifically uses the environments where few probing states are needed to learn a policy.

Minor Comments:
- The "Policy fingerprinting" paragraph should be divided to more clearly partition novel contributions from existing work.
- By changing the rewards for Hopper, Walker, and Ant, those environments are no longer being solved (only modified versions of them). Likewise, gamma is part of the MDP; reporting total return (gamma=1) when gamma is 0.99 for only some evaluated methods is incorrect.
- Sec 4.3: "trough" -> "through"
- The limitation of the proposed method to deterministic policies is used to motivate baseline selection, but this limitation is a choice rather than part of the problem setup. To use this line of reasoning, one should also express why deterministic policies are advantageous.

Questions:
- How were domains selected for evaluation? Why are only two environments used for Fig 10?
- What is the behavior of your method with stochastic policies?
- Can your approach perform well when the policies have different architectures during training?
- Can vanilla PSSVF be used with weighted sampling?

**Summary Of The Paper:**

This work combines two existing approaches: representing a policy based on its behavior in a set of probing states and finding a successful policy via a critic that estimates the return of any input policy.

The ability to automatically learn probing states is shown via MNIST.
The performance on a few continuous control tasks is demonstrated.
The ability to use a trained critic to find a successful linear policy is also shown.


**Summary Of The Review:**

The proposed method is motivated by the difficulty scaling a policy-parameter-based approach. However, this benefit is not supported.
Additionally, the overall experiments are limited without sufficient justification.

---

> ### Author Response · Authors · 2022-11-15
> **Response to reviewer xK3x with comments and improvements [2/2]**
>
> > How were domains selected for evaluation? Why are only two environments used for Fig 10?
>
> We used the standard MuJoCo suite for evaluation, which is a widely used benchmark for continuous control. In Figure 10 we compare our algorithm with vanilla PSSVF. We selected the 2 most complex environments used in the vanilla PBVF paper, using their best found hyperparameters for comparison.
>
> > What is the behavior of your method with stochastic policies?
>
> With stochastic policies our method would need to learn many more parameters since, e.g. with a Gaussian policy, we would have to concatenate means and standard deviations in the probing actions. This could result in a less sample efficient algorithm. Moreover, with a stochastic policy the variance of the return would be higher. This is why evolutionary approaches (and our method), which must estimate a fitness from data, consider in general deterministic policies.
>
>
> > Can your approach perform well when the policies have different architectures during training?
>
> In principle, as long as the different architectures have capacity to solve the task, our method can perform well since the value function is invariant to changes policy architecture. One challenge with having different architectures might be that the different policies could require different levels of noise (for exploration) and learning rates.
>
> > Can vanilla PSSVF be used with weighted sampling?
>
> Yes, vanilla PSSVF could benefit from such a trick. However, vanilla PSSVF is not sound to be used with neural networks, therefore a direct comparison with it is unfeasible.
>
> > In addition, the experiments are oddly limited:
> only continuous control problems are used for main experiments
> only deterministic policies are used for main experiments
> only environments where few crucial states are needed are used for evaluation To support the claim that probing states are a better representation, thorough comparisons between different representations should be performed. In particular, to demonstrate that probing states scale better, evaluations should be performed with more complex environments.
>
> We mentioned in the previous point why we chose deterministic policies. Our experiments involve most of the environments in the MuJoCo suite and are more complex than the previous experiments in the PBVF and PVN papers.
>
> Note that the fact that few crucial states are needed for evaluation is something that we discovered afterwards, when testing our method through the experiments in Fig 4, 13, 14, 15, 16. We did not choose our environments because we knew they require few crucial states, which is why we focus on this surprising finding.
>
> Note that we are not claiming that ours is the best policy representation. We are simply combining two existing methods and scaling them up to much bigger policies and more complex environments than those that were previously within reach.
> Perhaps the reviewer could clarify which additional (complex) environments and representations they are referring to?
>
> > The prioritization of content inclusion needs to be improved. Key experiments are in the appendix while lots of content in the main body can be cut or moved to the appendix.
>
> > Sec 2 and 3 contain lots of non-novel content that is not needed within this work.
>
> > Evaluating the contribution of policy fingerprinting is crucial, yet it is in the appendix.
>
> We thank the reviewer for this suggestion. We have removed part of Section 2 from the main text. We have moved the ablation on the effect of the number of probing states in the main part of the paper. We welcome any further suggestions regarding which experiments the reviewer would like to see in the main paper.
>
> > Likewise, "a demonstration that fingerprinting can learn interesting states in MNIST" is of limited interest (no hypothesis is being tested; no baselines are used for comparison).
>
> The goal of the experiment in Figures 2 and 6 is to visualize learned probing states. The goal of the experiment in Figure 7 is to show extrapolation (the baseline here is the $12\%$ maximum accuracy in the training set). If the reviewer suggests another baseline to be used, we would be happy to consider it.
>
> > Also, learning a near-optimal linear policy does not show that this method has an advantage over directly using policy parameters (i.e., this ability is due to PBVF).
>
> We would like to clarify that the goal of this experiment is to show that our method has some invariances to policy architecture and not to show an advantage over vanilla PBVF.

---

> ### Author Response · Authors · 2022-11-15
> **Response to reviewer xK3x with comments and improvements [1/2]**
>
> We thank the reviewer for taking the time to read our Appendix and for providing very detailed feedback. Below we address the questions raised by the reviewer:
>
> > The authors claim that "Flattening the policy parameters ... is difficult to scale to larger policies," but do not show the new method scaling.
>
> We believe that this is evident based on the observation that the memory and compuational cost of Vanilla PSSVF (with flattened policy parameters) grows with larger policies. For instance, the neural network policy that we use (70K parameters), would require learning a value function with 17M parameters. Whereas our value function with policy fingerprinting is instead invariant to policy size and contains roughly 170k learnable parameters.
>
> A direct comparison between our method and vanilla PSSVF is therefore meaningless for large policies as vanilla PSSVF does not scale to larger policies by the algorithm's definition.
>
>
> > Unfortunately, the proposed method is not shown to scale to larger domains. Empirical evaluation is required to confirm favorable scaling, but all of the domains considered can be solved with simple policies. I am skeptical that this method scales better than using the policy parameters: a strength of policy gradient methods is that the policy complexity generally scales more slowly than the environment complexity. However, this method requires coverage of the state space, so the number of required crucial states may scale quickly. The key experiment is shown in the appendix (Fig 10) and is quite limited in scope. It specifically uses the environments where few probing states are needed to learn a policy.
>
> The proposed method is shown to scale to larger domains with respect to those used in PBVF and PVN papers. The former scaled PSSVF up to Swimmer and Hopper environments, while the latter was tested in an instance of CartPole and Swimmer. Note that Ant, Walker2d, HalfCheetah have larger state and action spaces.
>
> Nevertheless, the main contribution here is to make vanilla PSSVF capable of handling larger policies. Applying vanilla PSSVF to big neural networks is simply infeasible.
>
> Perhaps the reviewer could clarify which larger domains they are referring to?
>
>
> > The "Policy fingerprinting" paragraph should be divided to more clearly partition novel contributions from existing work.
>
>
> We thank the reviewer for suggesting this. The "policy fingerprinting" paragraph introduces the policy fingerprinting as in PVN and contextualizes it in the PSSVF case (where the only main distinction is the algorithm being applied in the online setting). As we mention in the paper, our method can be considered as the PVN algorithm applied to the online setting or the PSSVF algorithms that uses policy fingerprinting.  We believe that explaining these concept in details as we did in Section 3 is important to understand how they can be combined.
>
>
> > By changing the rewards for Hopper, Walker, and Ant, those environments are no longer being solved (only modified versions of them). Likewise, gamma is part of the MDP; reporting total return (gamma=1) when gamma is 0.99 for only some evaluated methods is incorrect.
>
> We politely disagree with the reviewer on these points. Most RL algorithms are using some sort of reward shaping and it is perfectly fine to still evaluate them on the cumulative reward. In our specific case, removing the survival bonus from the environment does not change the optimal policies. Regarding $\gamma$ being part of the MDP, we would like to ask the reviewer how they would compare in an MDP with $\gamma=1$, an evolutionary method and DDPG (which must use $\gamma<1$). Is the reviewer suggesting that we should not use DDPG or SAC to solve MuJoCo environments because the have $\gamma=1$?
>
> > The limitation of the proposed method to deterministic policies is used to motivate baseline selection, but this limitation is a choice rather than part of the problem setup. To use this line of reasoning, one should also express why deterministic policies are advantageous.
>
> We thank the reviewer for pointing this out. Since here we are considering MDPs, there is always a deterministic optimal policy, so in general considering deterministic policies does not restrict the expressiveness of the actor. The main motivation for using deterministic policies though, as in evolutionary methods, is that the stochasticity of the noise injected in parameter space is enough for exploration. In evolutionary approaches and in our method it is preferable to collect returns using a deterministic policy rather than a stochastic one, since the return from a deterministic policy has lower variance in general. Note that we compare our method also with stochastic SAC, although this comparison is mainly to show the gap between our method and the state-of-the-art for continuous control.

---

> ### Author Response · Authors · 2022-12-09
> **Friendly reminder**
>
> The discussion period will end soon and we are happy to provide additional comments and engage in further discussion if needed.
> If the reviewer is satisfied with our response, we would appreciate it if they would consider increasing the score for the submission.

---

### Author Response · Authors · 2022-11-15
**General response**

We thank all reviewers for their insightful comments. We reply to each of them individually below. We have updated the paper to reflect most of the changes discussed in our response to the reviewers. We are happy to further engage with the reviewers and address any remaining concerns.

---

### Decision · Program_Chairs · 2023-01-20

**Decision:**

Reject

**Justification For Why Not Higher Score:**

See the weaknesses listed above.

**Justification For Why Not Lower Score:**

This is the lowest score.

**Metareview: Summary, Strengths And Weaknesses:**

Strengths: The main result is that policies in mujoco environments can be represented by observing the actions at a small number of states. It is surprising that a strong policy can be learned by regressing on the actions at those few states. If probing states are a more efficient representation for complex policies, then this method is of interest.

Weaknesses: The authors but do not show that the new method scales.  All of the domains considered can be solved with simple policies. IA reviewer is skeptical that this method scales better than using the policy parameters. The key experiment is shown in the appendix (Fig 10) and is quite limited in scope. It specifically uses the environments where few probing states are needed to learn a policy.

Also, the novelty is incremental. The proposed algorithm is a combination of the architecture of policy evaluation networks and online algorithm structure of parameter-based value functions.

Overall the reviewers did find sufficient merits in this paper.